# Visualizing the metazoan proliferation-quiescence decision in vivo

Rebecca C Adikes[1†], Abraham Q Kohrman[1†], Michael A Q Martinez[1†], Nicholas J Palmisano[1], Jayson J Smith[1], Taylor N Medwig-Kinney[1], Mingwei Min[2], Maria D Sallee[3], Ononnah B Ahmed[1], Nuri Kim[1], Simeiyun Liu[1‡], Robert D Morabito[1], Nicholas Weeks[1], Qinyun Zhao[1], Wan Zhang[1], Jessica L Feldman[3], Michalis Barkoulas[4], Ariel M Pani[5], Sabrina L Spencer[2], Benjamin L Martin[1], David Q Matus[1*]

[1]Department of Biochemistry and Cell Biology, Stony Brook University, Stony Brook, United States; [2]Department of Biochemistry and BioFrontiers Institute, University of Colorado Boulder, Boulder, United States; [3]Department of Biology, Stanford University, Stanford, United States; [4]Department of Life Sciences, Imperial College, London, United Kingdom; [5]Department of Biology, University of Virginia, Charlottesville, United States

*For correspondence:
david.matus@stonybrook.edu

[†]These authors contributed equally to this work

Present address: [‡] Molecular, Cell and Developmental Biology, University of California Santa Cruz, Santa Cruz, United States

Competing interests: The authors declare that no competing interests exist.

**Abstract** Cell proliferation and quiescence are intimately coordinated during metazoan development. Here, we adapt a cyclin-dependent kinase (CDK) sensor to uncouple these key events of the cell cycle in *Caenorhabditis elegans* and zebrafish through live-cell imaging. The CDK sensor consists of a fluorescently tagged CDK substrate that steadily translocates from the nucleus to the cytoplasm in response to increasing CDK activity and consequent sensor phosphorylation. We show that the CDK sensor can distinguish cycling cells in G1 from quiescent cells in G0, revealing a possible commitment point and a cryptic stochasticity in an otherwise invariant *C. elegans* cell lineage. Finally, we derive a predictive model of future proliferation behavior in *C. elegans* based on a snapshot of CDK activity in newly born cells. Thus, we introduce a live-cell imaging tool to facilitate in vivo studies of cell-cycle control in a wide-range of developmental contexts.

## Introduction

Organismal development requires a delicate balance between cell proliferation and cell cycle exit. In early embryos, the emphasis is placed on rapid cell proliferation, which is achieved by omitting gap phases (G1 and G2) and establishing a biphasic cell cycle that rapidly alternates between DNA synthesis (S phase) and mitosis (M phase) (*Edgar and O'Farrell, 1989*; *Newport and Kirschner, 1982*). After several rounds of embryonic cell division, the gap phases are introduced, coincident in many organisms with cell fate decisions and the execution of morphogenetic cell behaviors (*Foe, 1989*; *Grosshans and Wieschaus, 2000*). These gap phases are believed to function as commitment points for cell-cycle progression decisions. The earliest point of commitment occurs during G1, which is the focus of this study. Cells either engage in cell-cycle progression and enter S phase, or they exit the cell cycle altogether and enter a cell-cycle phase referred to as G0 and undergo quiescence or terminal differentiation (*Sun and Buttitta, 2017*). Although the location of the G1 commitment point in yeast (Start) and cultured mammalian cells (Restriction Point) has in large part been spatiotemporally mapped and molecularly characterized (*Hartwell et al., 1974*; *Pardee, 1974*; *Spencer et al., 2013*), when cells make this decision in living organisms while integrating intrinsic and the extrinsic cues of their local microenvironment during development remains poorly understood. A cell-cycle sensor

**eLife digest** All living things are made up of cells that form the different tissues, organs and structures of an organism. The human body, for example, is thought to consist of some 37 trillion cells and harbor over 200 cell types. To maintain a working organism, cells divide to create new cells and replace the ones that have died.

Cell division is a tightly controlled process consisting of several steps, and cells continuously face a Shakespearean dilemma of deciding whether to continue dividing (also known as cell proliferation) or to halt the process (known as quiescence). This difficult balancing act is critical during all stages of life, from embryonic development to tissue growth in an adult. Problems in the underlying pathways can result in diseases such as cancer.

Cell division is driven by proteins called CDKs, which help cells to complete their cell cycle in the correct sequence. To gain more insight into this complex process, scientists have developed tools for monitoring CDKs. One such tool is a fluorescent biosensor, a molecule that can be inserted into cells that glows and moves in response to CDK activity. The biosensor can be studied and measured in each cell using a microscope.

Adikes, Kohrman, Martinez et al. adapted and optimized an existing CDK biosensor to help study cell division and the switch between proliferation and quiescence in two common research organisms, the nematode *Caenorhabditis elegans* and the zebrafish. Analysis of this biosensor showed that CDK activity at the end of cell division is higher if the cells will divide again but is low if the cells are going to become quiescent. This could suggest that the decision of a cell between proliferation and quiescence may happen earlier than expected. The optimized biosensor is sensitive enough to detect these differences and can even measure variations that influence proliferation in a region on *C. elegans* that was once thought to be unchanging.

The development of this biosensor provides a useful research tool that could be used in other living organisms. Many research questions relate to cell division and so the applications of this tool are wide ranging.

---

that is amenable to such in vivo studies can shed new light on this four-decade-old biological phenomenon.

In 2008, Sakaue-Sawano and colleagues engineered a multicolor fluorescent ubiquitination-based cell-cycle indicator (FUCCI) for mammalian cell culture (*Sakaue-Sawano et al., 2008*). FUCCI has since been adapted for many research organisms (*Özpolat et al., 2017*; *Zielke and Edgar, 2015*). However, FUCCI on its own cannot distinguish between a cell residing in G1 that will cycle again upon completing mitosis and a cell that is poised to enter G0 (*Oki et al., 2015*). Separating G1 from G0 is an essential first step to understanding mechanisms controlling cell cycle exit during quiescence or terminal differentiation. To distinguish G1 from G0 in mammalian cell culture, Hahn, Spencer and colleagues developed and implemented a single-color ratiometric sensor of cell-cycle state composed of a fragment of human DNA helicase B (DHB) fused to a fluorescent protein that is phosphorylated by CDKs (*Hahn et al., 2009*; *Schwarz et al., 2018*; *Spencer et al., 2013*). Notably, through quantitative measurements of CDK activity, this sensor provided new insights into the proliferation-quiescence decision in cultured mammalian cells by identifying cycling cells that exit mitosis in a CDK-increasing (CDK$^{inc}$) state and quiescent cells that exit mitosis in a CDK-low (CDK$^{low}$) state (*Spencer et al., 2013*). Nonetheless, a DHB-based CDK sensor has not been utilized to evaluate the proliferation-quiescence decision in vivo.

In this study, we investigate the proliferation-quiescence decision in *Caenorhabditis elegans* and zebrafish, two powerful in vivo systems with radically different modes of development. We generate transgenic CDK sensor lines in each organism to examine this decision live at mitotic exit. By quantifying CDK activity, or DHB ratios, at mitotic exit, we are able to predict future cell behavior across several embryonic and post-embryonic lineages. Despite cells generally exiting mitosis with decreased CDK-activity levels, we reliably distinguish cycling cells that exit mitosis into G1, in a CDK$^{inc}$ state, from quiescent cells that exit mitosis into G0, in a CDK$^{low}$ state. To gain insights into cell-cycle progression commitment, we examine the activity of *C. elegans cki-1*, a cyclin-dependent kinase inhibitor (CKI) of the Cip/Kip family, demonstrating that endogenous CKI-1 levels are anti-

correlated with CDK activity during the proliferation-quiescence decision. We propose that integration of CKI-1 levels in the mother cell and high CKI-1, low CDK activity at mitotic exit mediate this decision. By utilizing the CDK sensor to predict future cell behavior, we uncover a cryptic stochasticity that occurs in a temperature-dependent fashion in the *C. elegans* vulva, an otherwise invariant and well-characterized lineage. Finally, we reveal cell-cycle dynamics in zebrafish, an organism that lacks a defined cell lineage, demonstrating that quiescent embryonic tissues display DHB ratios that correlate with those observed in G0 cells in *C. elegans*. Together, we present a tool for visualizing G1/G0 dynamics in vivo during metazoan development that can be used to study the interplay between cell proliferation and quiescence.

## Results

### Design and characterization of a live *C. elegans* CDK sensor to define interphase states

We synthesized a *C. elegans* codon-optimized fragment of human DHB composed of amino acids 994–1087 (*Hahn et al., 2009*; *Spencer et al., 2013*). The fragment contains four serine residues that are consensus CDK phosphorylation sites (*Moser et al., 2018*; *Spencer et al., 2013*). These serines flank a nuclear localization signal (NLS) that is adjacent to a nuclear export signal (NES) (*Figure 1A*). When CDK activity is low, the NLS is strong, the NES is weak and DHB localizes to the nucleus. However, when CDK activity increases during cell-cycle entry, the NLS is masked and DHB re-localizes to the cytoplasm (*Figure 1B*). Using this DHB fragment, we generated two CDK sensors by fusing green fluorescent protein (GFP) or two copies of a red fluorescent protein, mKate2 (2xmKate2), to the DHB C-terminus (*Figure 1A*). To visualize the nucleus, we co-expressed *his-58*/histone H2B fused to 2xmKate2 or GFP, respectively, which is separated from DHB by a P2A self-cleaving viral peptide (*Ahier and Jarriault, 2014*). We drove the expression of each CDK sensor via a ubiquitous *rps-27* promoter (*Ruijtenberg and van den Heuvel, 2015*).

To test both the GFP (*Figure 1—figure supplement 1A and B*) and 2xmKate2 (*Figure 1C*, *Figure 1—figure supplement 1B and C*) versions of our CDK sensor, we began by examining cell divisions in the *C. elegans* embryo and germline (*Figure 1—video 1*). First, we visualized cells in the embryonic intestine, which is clonally derived from the E blastomere, as these are the first cells in the embryo to have gap phases (*Edgar and McGhee, 1988*). The E blastomere goes through four rounds of divisions to give rise to 16 descendants (E16 cells) about 4 hr after first cleavage. While 12 of the E16 cells have completed their embryonic divisions at this stage (*Leung et al., 1999*), four cells called E16* star cells divide once more to generate the 20-celled intestine (E20) (*Rasmussen et al., 2013*; *Yang and Feldman, 2015*). Thus, we wondered whether our CDK sensor could be used to distinguish between cycling E16* star cells and quiescent E16 cells. To accomplish this, we tracked E16* star cell division from the E16–E20 stage and observed that DHB::GFP localizes in a cell-cycle-dependent fashion during these divisions, with DHB::GFP translocating from the nucleus to the cytoplasm and then re-locating to the nucleus at the completion of E16* star cell division (*Figure 1—figure supplement 1A*). Consistent with our observations using the GFP version of our CDK sensor in mid-embryogenesis, DHB::2xmKate2 also dynamically translocates from the nucleus to the cytoplasm during cell divisions in the early embryo (*Figure 1C*). Second, we examined the localization of DHB::GFP and DHB::2xmKate2 in the adult *C. elegans* germline (*Figure 1—figure supplement 1B and C*). Here we detected a gradient of live CDK activity, from high in the distal mitotic progenitor zone to low in the proximal meiotic regions, as described with EdU incorporation and phospho-histone H3 staining (*Kocsisova et al., 2018*). Together, these results demonstrate that our CDK sensor is dynamic during cell-cycle progression in the *C. elegans* embryo and germline.

The ability to distinguish cycling cells from quiescent cells in the embryo made us wonder whether we could also distinguish these cellular states post-embryogenesis. Therefore, we examined our CDK sensor in several post-embryonic somatic lineages that undergo proliferation followed by cell cycle exit (*Sulston and Horvitz, 1977*). Specifically, we selected the sex myoblasts (SM), the somatic sheath (SS) and ventral uterine (VU) cells of the somatic gonad, and the vulval precursor cells (VPCs) (*Figure 1D and D'*). To define each phase of the cell cycle while these lineages are proliferating, we combined static and time-lapse imaging approaches to measure cytoplasmic:nuclear DHB ratios for G1, S, and G2 (*Figure 1E–J*, *Figure 1—figure supplement 1D–M*). First, we quantified DHB ratios

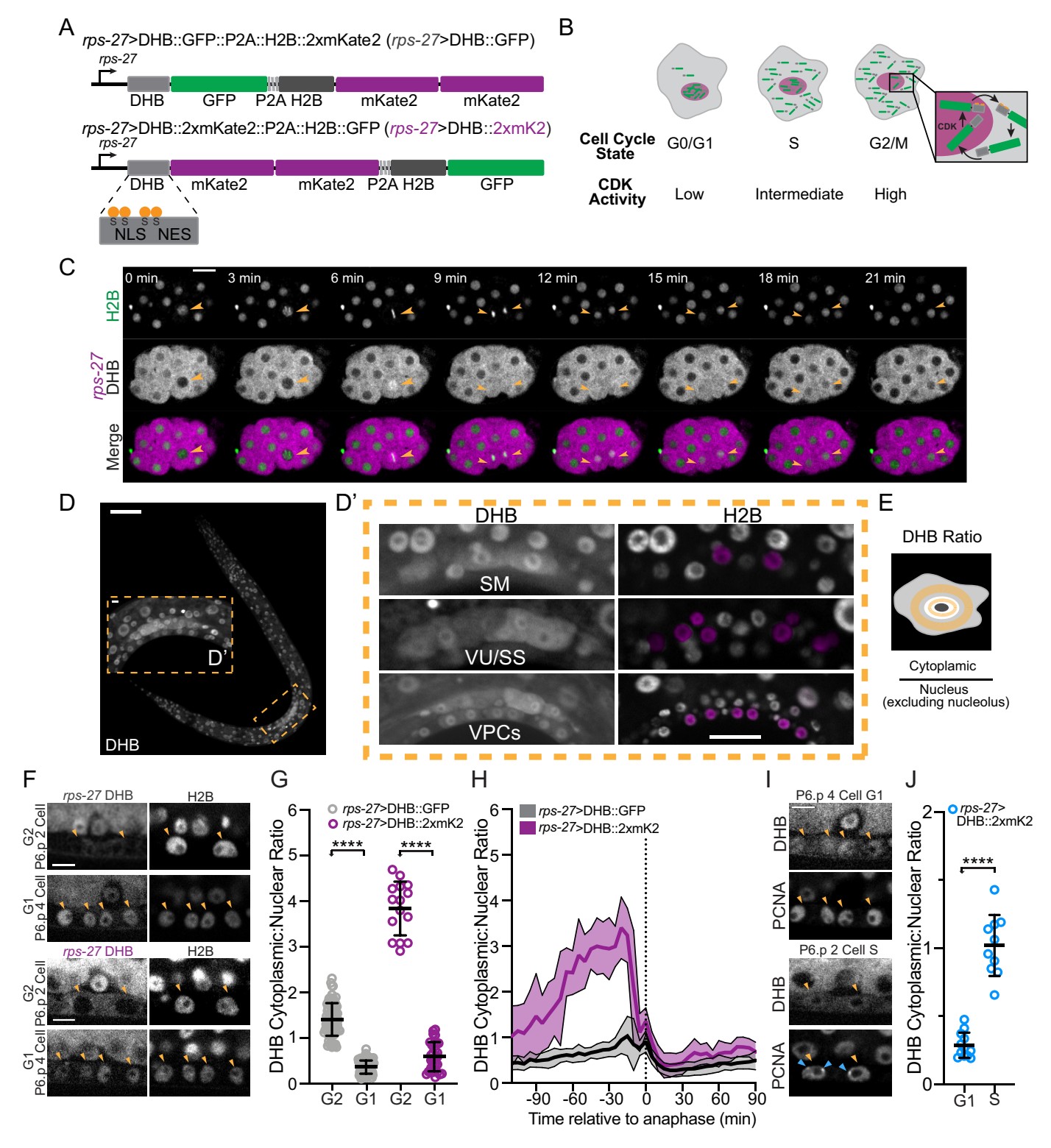

**Figure 1.** Design and characterization of a live *C. elegans* CDK sensor to define interphase states. (**A**) Schematic of the CDK sensor fused to GFP (top) or 2xmKate2 (bottom) and a nuclear mask (H2B::FP) separated by a self-cleaving peptide (P2A). Inset: nuclear localization signal (NLS), nuclear export signal (NES), and consensus CDK phosphorylation sites on serine (S) residues. (**B**) Schematic of CDK sensor translocation during the cell cycle. (**C**) Representative fluorescence overlay (bottom), H2B (top), and DHB::2xmKate2 (middle) time series images during embryo cell divisions (see ***Figure 1—video 1***). Orange arrowheads follow the division of a single blastomere. (**D**) Confocal micrograph montage of CDK sensor expression in a *C. elegans* L3

*Figure 1 continued on next page*

*Figure 1 continued*

stage larva. (**D′**) Three somatic tissues are highlighted (inset, dashed orange box) shown at higher magnification with pseudo-colored nuclei (magenta) depicting cells of interest. (**E**) Schematic of quantification and equation used to obtain the cytoplasmic:nuclear ratio of DHB. (**F**) Representative images of sensor expression in vulval precursor cells (VPCs) at peak G2 and 20 min after anaphase during G1 in DHB::GFP (gray) and DHB::2xmKate2 (magenta). Orange arrowheads indicate the VPCs. (**G**) Dot plot depicting G2 and G1 DHB ratios of the two CDK sensor variants in the VPCs ($n \geq 15$ cells per phase). (**H**) Plot of DHB ratios in VPCs during one round of cell division, measured every 5 min ($n \geq 11$ mother cells per strain). Dotted line indicates time of anaphase. Error bars and shaded error bands depict mean ± SD. (**I**) Representative images of sensor and PCNA expression in VPCs during G1 and S phase. Orange arrowheads indicate the VPCs. Blue arrowheads indicate S phase PCNA puncta. (**J**) Dot plot depicting G1 and S phase DHB::2xmKate2 ratios based on absence or presence of PCNA puncta ($n \geq 10$ cells per phase). \*\*$p \leq 0.01$, \*\*\*\*$p \leq 0.0001$. Significance determined by statistical simulation; *p*-values in ***Supplementary file 1***. Scale bar = 10 µm (except in D: 20 µm and F, I: 5 µm).

The online version of this article includes the following video, source data, and figure supplement(s) for figure 1:

**Source data 1.** Source data for *Figure 1*.
**Source data 2.** Source data for *Figure 1—figure supplement 1*.
**Figure supplement 1.** Visualization of CDK activity live in embryonic and post-embryonic tissues.
**Figure 1—video 1.** Representative time-lapse of *C. elegans* germline and embryo expressing rps-27>DHB::2x-mKate2, related to *Figure 1*, *Figure 1— figure supplement 1*.
https://elifesciences.org/articles/63265#fig1video1
**Figure 1—video 2.** Representative time-lapse of S to G2 transition during sex myoblast (SM) division expressing *rps-27*>DHB::2xmKate2 and *pcn-1*>PCN-1::GFP, Related to *Figure 1*, *Figure 1—figure supplement 1*.
https://elifesciences.org/articles/63265#fig1video2

following time-lapse of VPC (*Figure 1F–H*), SM (*Figure 1—figure supplement 1D, F and G*) and uterine (*Figure 1—figure supplement 1E, F and H*) divisions to determine peak values of G2 CDK activity. All lineages exhibited the same CDK sensor localization pattern during peak G2 (i.e. maximal nuclear exclusion). We then RNAi depleted the sole *C. elegans* CDK1 homolog, *cdk-1,* to induce a penetrant G2 phase arrest in the SM cells to corroborate these results. Quantification of DHB ratios following *cdk-1* RNAi treatment showed a mean ratio of 1.00 ± 0.28 and 2.36 ± 0.70 in the GFP and 2xmKate2 versions of our CDK sensor, respectively (*Figure 1—figure supplement 1I and J*). Next, for each lineage (*Figure 1F*, *Figure 1—figure supplement 1D and E*), we quantified DHB ratios 25 min after anaphase from our time-lapses to determine a threshold for G1 phase CDK activity. In G1, DHB::GFP and DHB::2xmKate2 were nuclear localized after mitotic exit with mean ratios of 0.35 ± 0.14 and 0.58 ± 0.32 in VPCs, 0.59 ± 0.11 and 0.97 ± 0.20 in SMs, and 0.67 ± 0.10 and 1.13 ± 0.17 in uterine cells (*Figure 1G and H*, *Figure 1—figure supplement 1F–H*). Finally, we paired DHB::2xmKate2 with a reporter for S phase, fusing GFP to the sole *C. elegans* proliferating cell nuclear antigen (PCNA) homolog, *pcn-1*, expressed under its own endogenous promoter at single copy. Although nuclear localized throughout the cell cycle, PCNA forms sub-nuclear puncta only in S phase (*Brauchle et al., 2003*; *Dwivedi et al., 2019*; *Strzyz et al., 2015*; *Zerjatke et al., 2017*). Analysis of time-lapse data found that punctate expression of PCN-1::GFP correlated with mean DHB::2xmKate2 ratios of 1.02 ± 0.22 in VPC (*Figure 1I and J*), 0.89 ± 0.16 in SM (*Figure 1—figure supplement 1K and M*; *Figure 1—video 2*), and 1.00 ± 0.10 in uterine (*Figure 1—figure supplement 1L and M*) lineages. Despite individual lineages showing differences in CDK activity (*Figure 1—figure supplement 1F and M–O*), primarily in G1, we can establish DHB ratios for each interphase state (G1/S/G2) across several post-embryonic somatic lineages using our CDK sensor paired with a PCNA reporter. We next wondered if we could distinguish G1 from G0 as these somatic lineages exit their final cell division; therefore, allowing us to visibly and quantitatively detect cellular quiescence in vivo. We mainly chose the DHB::GFP version of our CDK sensor to conduct the following experiments as it was more photostable.

## CDK^low^ activity after mitotic exit is predictive of cell cycle exit

In asynchronously dividing MCF10A epithelial cell lines, cells that exited mitosis into a CDK2^low^ state had a high probability of staying in G0 compared to cells that exited at a CDK2^inc^ state (*Spencer et al., 2013*). We therefore wanted to determine whether the cytoplasmic:nuclear ratio of DHB::GFP following an in vivo cell division could be used to predict if a cell will enter G1 and divide again or enter G0 and undergo quiescence. Taking advantage of the predictable cell lineage pattern of *C. elegans*, we quantitatively correlated DHB::GFP ratios with the decision to proliferate or exit

the cell cycle. We first quantified DHB::GFP ratios from time-lapse acquisitions of SM cell divisions. The SM cells undergo three rounds of cell division during the L3 and L4 larval stages before exiting the cell cycle and differentiating into uterine muscle (um) and vulval muscle (vm) (*Figure 2A*; *Sulston and Horvitz, 1977*). Quantification of DHB::GFP in this lineage revealed that shortly after the first and second divisions, CDK activity increases immediately after mitotic exit from an intermediate level, which we designate as a CDK$^{inc}$ state (*Figure 2B and C*; *Figure 2—video 1*), Conversely, CDK activity following the third and terminal division remains low, which we designate as a CDK$^{low}$ state. Bootstrap analyses support a significant difference in DHB::GFP ratios between pre-terminal (CDK$^{inc}$) and terminal divisions (CDK$^{low}$), but not among pre-terminal divisions (*Figure 2—figure supplement 1A–C*). We then quantified DHB::GFP ratios during the division of two somatic gonad lineages, the VU and SS cells. VU and SS cells undergo several rounds of division during the L3 larval stage and exit the cell cycle in the early L4 stage (*Figure 2D*; *Sulston and Horvitz, 1977*). We quantified a pre-terminal division and the subsequent division that leads to quiescence. Similar to the SM lineage, both somatic gonad lineages exit the round of cell division prior to their final division into a CDK$^{inc}$ state and then exit into a CDK$^{low}$ state following their terminal division (*Figure 2E and F*, *Figure 2—figure supplement 1D–F*; *Figure 2—video 2*). Bootstrap analyses also support a significant difference between DHB::GFP ratios in pre-terminal versus terminal divisions in the developing somatic gonad (*Figure 2—figure supplement 1D*).

Next, we sought to determine how the CDK sensor behaves under conditions in which cells are experimentally forced into G0. To accomplish this, we generated a single copy transgenic line of mTagBFP2-tagged CKI-1, the *C. elegans* homolog of p21$^{Cip1}$/p27$^{Kip1}$, under an inducible heat shock promoter (*hsp*), paired with a *rps-0*>DHB::mKate2 variant of the CDK sensor. Induced expression of CKI-1 is expected to result in G0 arrest (*Hong et al., 1998*; *Matus et al., 2014*; *van der Horst et al., 2019*). Indeed, in the SM and uterine lineages, induced expression of CKI-1 resulted in cells entering a CDK$^{low}$ G0 state, with mean DHB ratios of 0.10 ± 0.05 and 0.12 ± 0.05, respectively (*Figure 2G*), as compared to control animals that lacked heat shock-induced expression (SM: 0.99 ± 0.82, uterine: 0.71 ± 0.35) or lacked the inducible transgene (SM: 0.96 ± 0.77, uterine: 1.00 ± 0.37). Thus, induced G0 arrest by ectopic expression of CKI-1 is functionally equivalent, by CDK-activity levels, to the G0 arrest that occurs following mitotic exit in an unperturbed cell destined to undergo quiescence.

We next examined the divisions of the 1°- and 2°-fated VPC lineage. The *C. elegans* vulva is derived from three cells (P5.p–P7.p), which undergo three rounds of cell division during the L3 and early L4 larval stages (*Figure 3A and B*; *Katz et al., 1995*; *Sternberg and Horvitz, 1986*; *Sulston and Horvitz, 1977*). Rather than giving rise to 24 cells, the two D cells, the innermost granddaughters of the 2°-fated P5.p and P7.p, exit the cell cycle one round early. This results in a total of 22 cells, which comprise the adult vulva (*Katz et al., 1995*; *Sulston and Horvitz, 1977*). Quantification of DHB::GFP ratios during VPC divisions yielded the expected pattern. The daughters of P5.p–P7.p all exited their first division into a CDK$^{inc}$ state (*Figure 3C and D*). After the next division, the 12 granddaughters of P5.p–P7.p (named A–F symmetrically) are born, including the D cell (*Katz et al., 1995*; *Sulston and Horvitz, 1977*). At this division, the strong nuclear localization of DHB::GFP in the D cell was in stark contrast to the remaining proliferating VPCs. The D cell exited into and remained in a CDK$^{low}$ state, while the remaining VPCs exited into a CDK$^{inc}$ state and continued to progress through the cell cycle (*Figure 3C and D*; *Figure 3—video 1*). All remaining VPCs exited into a CDK$^{low}$ state at their terminal division. Consistent with these results, bootstrap analyses (*Figure 3—figure supplement 1A–G*) support our qualitative results, such that we can quantitatively distinguish between a cell that has completed mitosis and will continue to cycle (CDK$^{inc}$) from a cell that exits mitosis and enters a G0 state (CDK$^{low}$).

## CKI-1 levels peak prior to cell cycle exit

In mammalian cell culture, endogenous levels of p21$^{Cip1}$ during G2 are predictive of whether a cell will go on to divide or enter quiescence, senescence, or terminal differentiation (*Hsu et al., 2019*; *Moser et al., 2018*; *Overton et al., 2014*; *Spencer et al., 2013*). This raises the intriguing possibility that endogenous levels of CKI-1 in *C. elegans* correlate with CDK$^{low}$ or CDK$^{inc}$ activity. To co-visualize CKI-1 dynamics with our CDK sensor, we inserted a N-terminal GFP tag into the endogenous locus of *cki-1* via CRISPR/Cas9 and introduced a DHB::2xmKate2 variant of the sensor (devoid of histone H2B) into this genetic background. Since endogenous levels of GFP::CKI-1 were too dim for

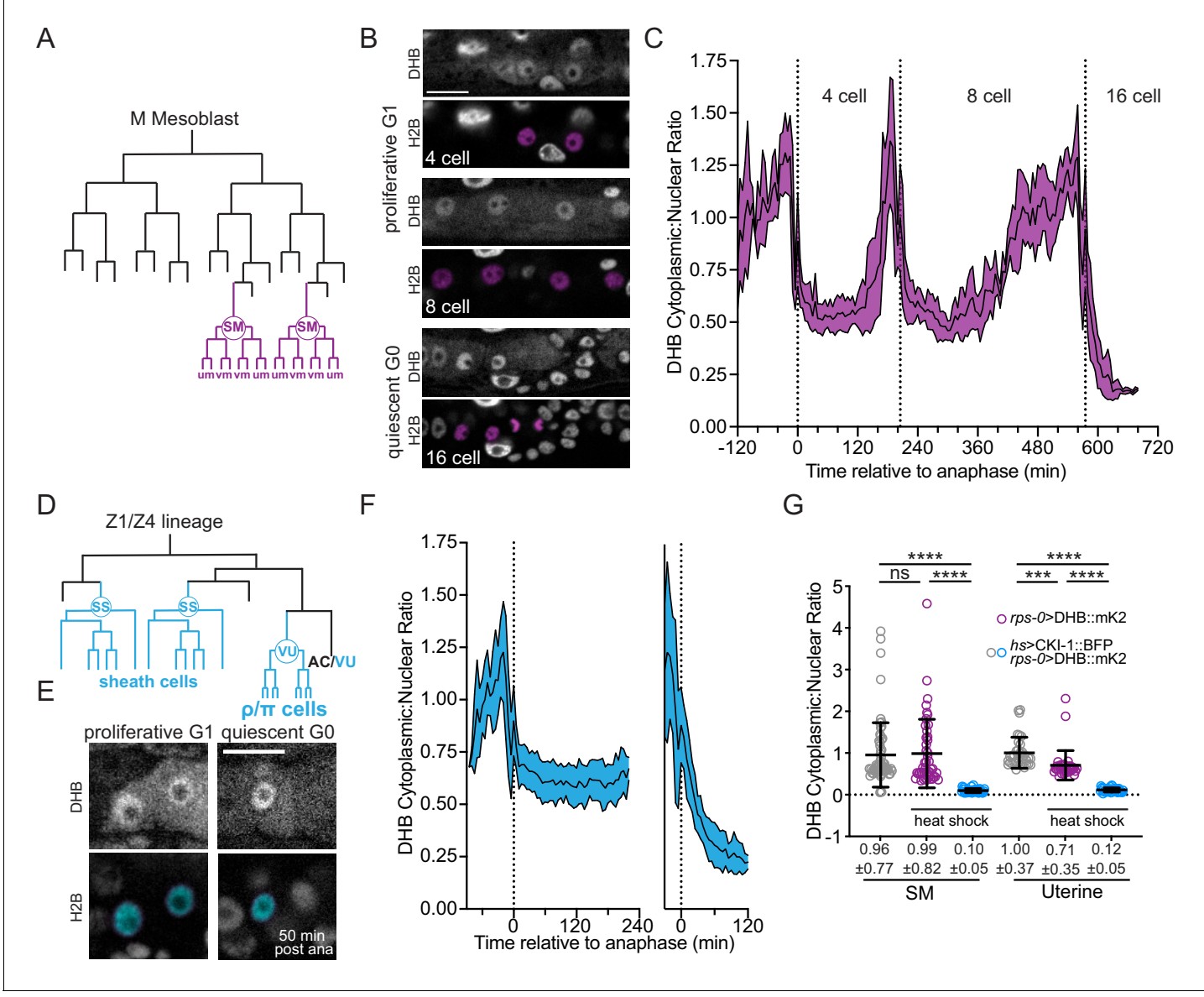

**Figure 2.** Sex myoblasts and somatic gonad cells exit terminal divisions into a CDK$^{low}$ state. (**A**) SM lineage schematic. (**B**) Micrographs of a time-lapse showing SM cells at G1 and G0. (**C**) Quantification of CDK activity in SM cells ($n \geq 10$). (**D**) Uterine lineage schematic. (**E**) Micrographs of a time-lapse showing uterine cells at G1 and G0. (**F**) Quantification of CDK activity in uterine cells ($n \geq 13$). (**G**) Quantification of CDK activity in SM cells and uterine cells following ectopic expression of CKI-1 (*hsp*>CKI-1:: mTagBFP2) compared to non-heat shock controls and heat shock animals without the inducible transgene ($n \geq 36$ cells per treatment). Pseudo-colored nuclei magenta, B; cyan, (**E**) indicate cells of interest. Scale bars = 10 μm. Dotted line in C and F indicates time of anaphase. Line and shaded error bands depict mean ± SD. Time series measured every 5 min. ns, not significant, ****p≤0.0001. Significance determined by statistical simulations; *p*-values in ***Supplementary file 1***.

The online version of this article includes the following video, source data, and figure supplement(s) for figure 2:

**Source data 1.** Source data for ***Figure 2***.

**Source data 2.** Source date for ***Figure 2—figure supplement 1***.

**Figure supplement 1.** CDK sensor can detect differences in proliferation potential of the SM and uterine postembryonic blast lineages.

**Figure 2—video 1.** Representative time-lapse of pre-terminal and terminal sex myoblast (SM) division expressing *rps-27>DHB::GFP*, Related to ***Figure 2***, ***Figure 2—figure supplement 1***.

https://elifesciences.org/articles/63265#fig2video1

**Figure 2—video 2.** Representative time-lapse of pre-terminal and terminal uterine sheath (SS) divisions expressing rps-27>DHB::GFP, Related to ***Figure 2***, ***Figure 2—figure supplement 1***.

https://elifesciences.org/articles/63265#fig2video2

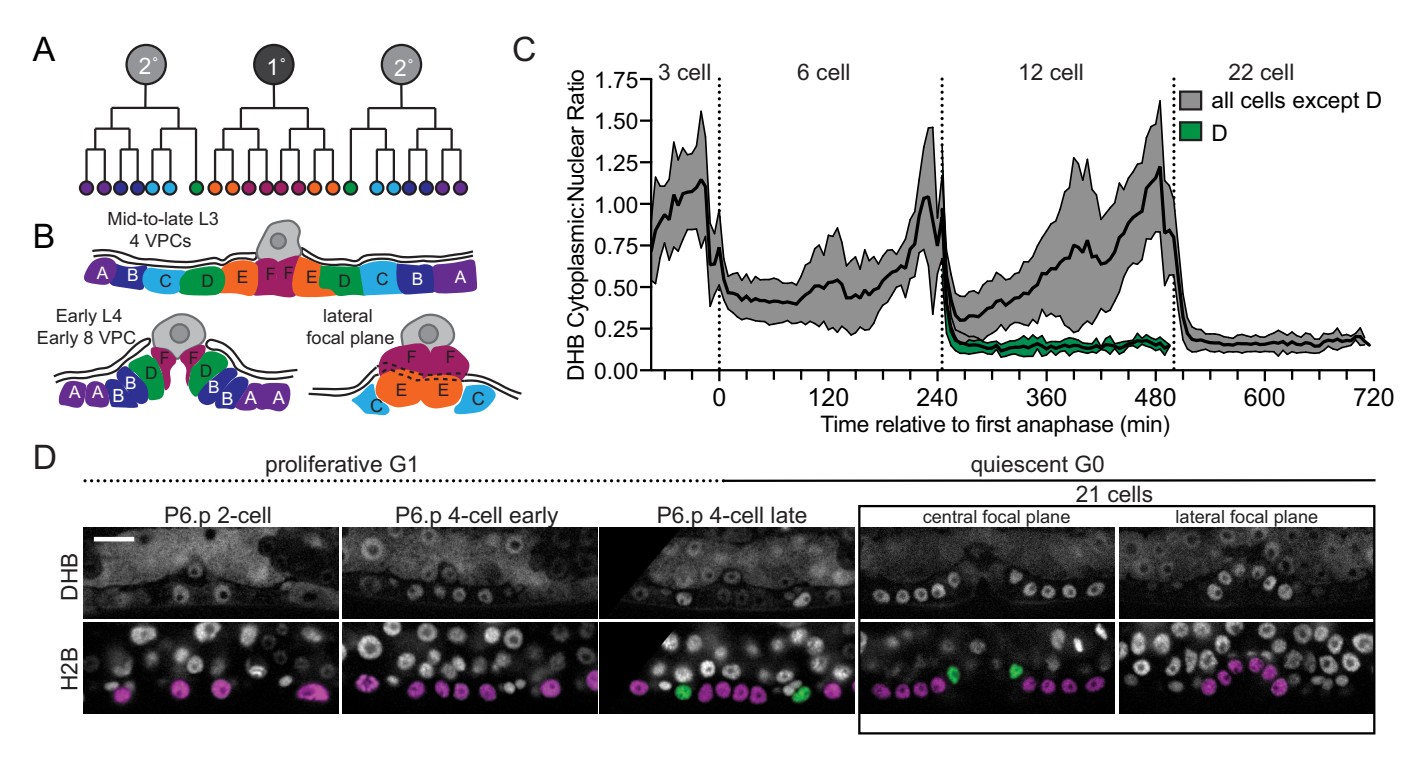

**Figure 3.** Vulval precursor cells (VPCs) exit terminal divisions into a CDK^low state. (**A**) Schematic of primary (1°) and secondary (2°) fated VPCs. (**B**) All of the VPCs divide, with the exception of the D cells, to facilitate vulval morphogenesis. (**C**) Time series of CDK sensor localization in the 1° and 2° VPCs, as measured every 5 min. Note that the D cells are born into a CDK^low state ($n \geq 9$ cells). Dotted line indicates time of anaphase. Shaded error bands depict mean ± SD. (**D**) Representative images of CDK sensor localization in the VPCs from the P6.p 2 cell stage to 8 cell stage. Nuclei (H2B) are highlighted in magenta for non-D cell 1° and 2° VPCs and green for the D cells. Scale bar = 10 μm.

The online version of this article includes the following video, source data, and figure supplement(s) for figure 3:

**Source data 1.** Source data for *Figure 3*.
**Source data 2.** Source date for *Figure 3—figure supplement 1*.
**Figure supplement 1.** CDK sensor can detect differences in proliferation potential of the VPC postembryonic blast lineage.
**Figure 3—video 1.** Representative time-lapse of pre-terminal and terminal vulval precursor cell (VPC) divisions expressing *rps-27>DHB::GFP*, Related to *Figure 3*, *Figure 3—figure supplement 1*.
https://elifesciences.org/articles/63265#fig3video1

time-lapse microscopy, likely due to its short half-life (*Yang et al., 2017*), we collected a developmental time series of static images over the L3 and L4 larval stages to characterize GFP::CKI-1 levels during pre-terminal and terminal divisions in the VPC lineage. We detected generally low levels of GFP::CKI-1 at the Pn.p 2 cell stage (*Figure 4A–C*, *Figure 4—figure supplement 1A–C*). In their daughter cells, at the Pn.p 4 cell stage, we detected an increase in GFP::CKI-1 levels in cycling cells prior to their next cell division, peaking in G2 (*Figure 4A, B and D*, *Figure 4—figure supplement 1A–C*). Notably, the D cell, which becomes post-mitotic after this cell division, exits mitosis with higher levels of GFP::CKI-1 than its CD mother (*Figure 4B*, *Figure 4—figure supplement 1B*). This trend holds true for the remaining VPCs at the Pn.p 6 cell and 8 cell stage, which show high levels of GFP::CKI-1 that peak immediately after mitotic exit and remain high during the post-mitotic L4 stage (*Figure 4A, B, E and F*, *Figure 4—figure supplement 1A–C*). We also observed increasing levels of GFP::CKI-1 in the G2 phase of mother cells that peak in their quiescent daughter cells in the uterine (*Figure 4—figure supplement 1D*) and SM cell lineages (*Figure 4—figure supplement 1E*). Thus, levels of GFP::CKI-1 increase in mother cells and remain high upon mitotic exit in daughter cells with CDK^low activity. These results suggest that the proliferation-quiescence decision is already underway in the G2 phase of the previous cell cycle and correlates with CKI-1 levels in the mother cell.

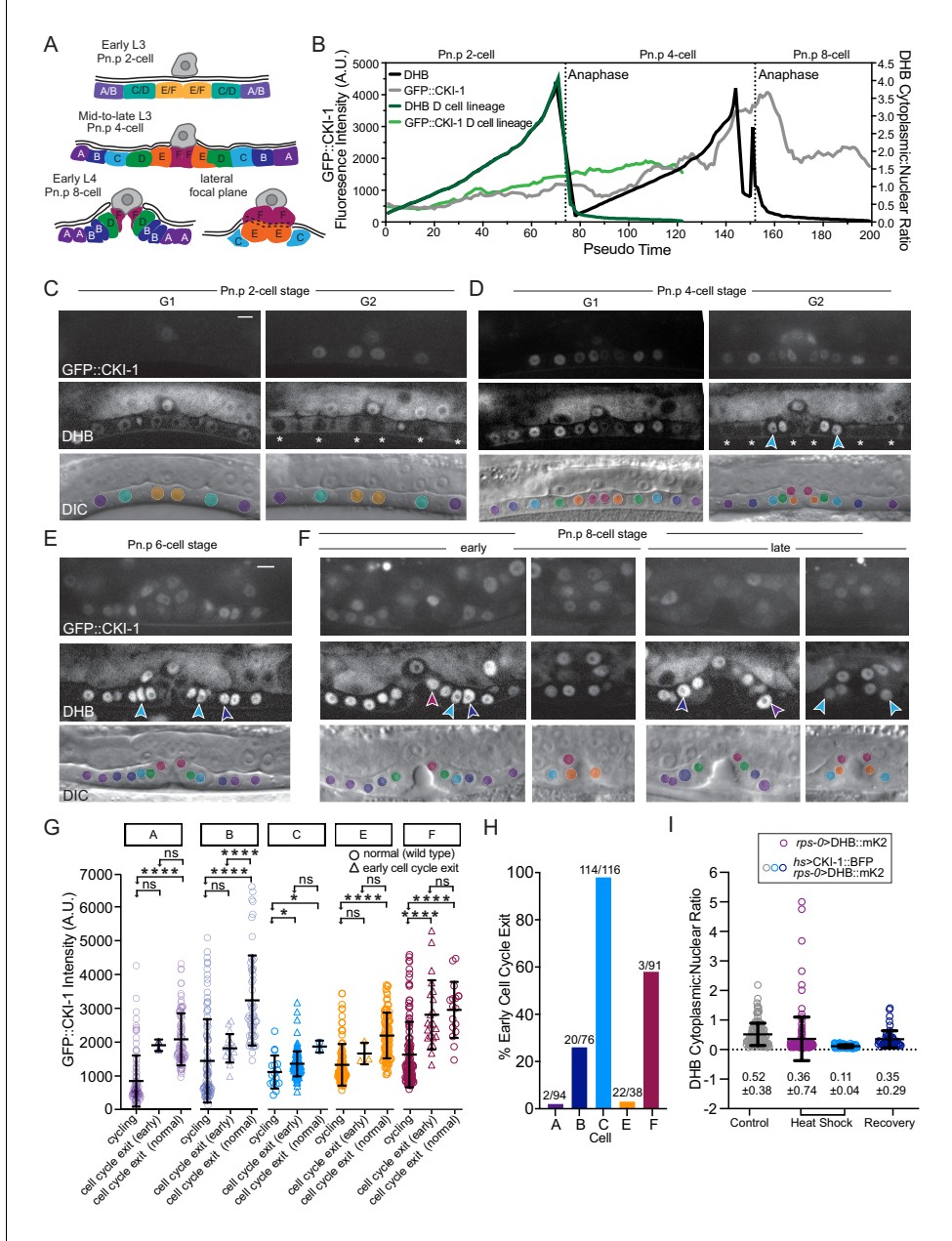

**Figure 4.** CKI-1 levels peak prior to cell cycle exit. (**A**) Schematic of VPC divisions in the L3-L4 larval stage. (**B**) CDK activity and CKI-1 levels across pseudo-time and DHB ratios for all VPCs (black line) and D cells (dark green line). GFP::CKI-1 fluorescence in VPCs (gray line) and D cell (light green line); $n \geq 93$ cells per lineage. (**C**) Representative images of VPCs at the Pn.p 2-cell stage at G1 and G2 (white asterisk). (**D**) Representative images of VPCs at the Pn.p 4-cell stage at G1 and G2; early quiescent C cells (cyan arrows) with low levels of GFP::CKI-1. (**E,** **F**) Representative images of VPCs at the Pn.p 6- cell stage (**E**) and 8-cell stage (**F**); arrows show early quiescent C (cyan) and B cell (dark blue), F cell (magenta), and A cell (purple). (**G**) GFP::CKI-1 fluorescence in each cell of the VPC lineage ($n \geq 16$, except C normal and A early $n = 2$, E early $n = 3$). (**H**) Percentage of cells of each lineage that showed signs of early quiescence and did not undergo their final division. (**I**) Overexpression of CKI-1 via heat shock causes cells to pre-maturely enter and remain in G0 ($n \geq 36$ cells per treatment). Scale bar = 10 µm. ns, not significant, *p≤0.05, ****p≤0.0001. Significance determined by statistical simulations; *p*-values in *Supplementary file 1*.

The online version of this article includes the following source data and figure supplement(s) for figure 4:

**Source data 1.** Source data for *Figure 4*.
**Figure supplement 1.** GFP::CKI-1 levels are predictive of future cell behavior.

During our collection of static images of GFP::CKI-1 animals, we observed significant deviations in the expected VPC lineage pattern in the early L4 larval stage. In particular, we noted that many cells appeared to bypass their final division and undergo early cell-cycle quiescence with coincident high levels of GFP::CKI-1 and low DHB ratios. We hypothesized that the line we generated could be behaving as a gain-of-function mutant, as GFP insertions at the N-terminus could interfere with proteasome-mediated protein degradation of CKI-1 (*Bloom et al., 2003*). The penetrance of this early cell-cycle quiescence defect varied across VPC lineages. While the A (2% of cases observed) and E (3% of cases observed) lineages showed a low penetrance of this defect, the B (26% of cases observed) and F (58% of cases observed) lineages showed a moderate penetrance (*Figure 4G and H*). We speculate that the A and E lineages are largely insensitive to the gain-of-function mutant because CKI-2, an understudied paralog of CKI-1, may be the dominant CKI in these cells. The C cell, sister to the D cell, had a highly penetrant early cell-cycle quiescence defect (98% of cases observed; *Figure 4G and H*). Consistent with our finding that high levels of endogenous GFP::CKI-1 can lead to early cell-cycle quiescence, heat shock-induced CKI-1 expression uniformly drove VPCs at the Pn.p 2 cell stage into a $CDK^{low}$ G0 state with mean DHB ratios of $0.11 \pm 0.05$ (*Figure 4I*), as compared to control animals that lacked heat shock-induced expression ($0.46 \pm 0.87$) or lacked the inducible *cki-1* transgene ($0.47 \pm 0.42$). Strikingly, most of the VPCs of heat shocked larvae that were allowed to recover for 5 hr remained quiescent ($0.35 \pm 0.29$) (*Figure 4I*), as VPCs that received an effective pulse of CKI-1 failed to divide again. We observed an average of 6.86 VPCs present hours later at the L4 stage as opposed to the wild type vulva composed of 22 total cells (*Figure 4—figure supplement 1F and G*). Together, these results demonstrate that cycling cells are highly sensitive to levels of CKIs and that increased expression can induce a G0 state.

## CDK activity predicts a cryptic stochastic fate decision in an invariant cell lineage

A strength of *C. elegans* is the organism's robust ability to buffer external and internal perturbations to maintain its invariant cell lineage. However, not all cell divisions that give rise to the 959 somatic cells are completely invariant. Studies have identified several lineages, including the vulva, where environmental stressors, genetic mutations and/or genetic divergence of wild isolates leads to stochastic changes in a highly invariant cell fate pattern (*Braendle and Félix, 2008*; *Hintze et al., 2020*; *Katsanos et al., 2017*). Thus, we wondered if the CDK sensor generated here could be utilized to visualize and predict stochastic lineage decisions during *C. elegans* development.

The VPC lineage that gives rise to the adult vulva is invariant under most conditions (*Figure 5A*, *Figure 5—figure supplement 1A*; *Sulston and Horvitz, 1977*). However, at high temperatures it has been observed that the D cell, the inner-most granddaughter of P5.p or P7.p, will go on to divide (*Figure 5A*; *Sternberg, 1984*; *Sternberg and Horvitz, 1986*). Unexpectedly, we noticed a rare occurrence of D cells expressing elevated DHB ratios during the course of time-lapse analysis of VPC divisions captured under standard laboratory conditions. To determine the penetrance of the cycling D cell phenotype, we inspected each of our CDK sensor lines grown at 25℃, a high temperature that is still within normal range for *C. elegans*. In both strains we observed a cycling D cell with a 4–6% penetrance (*Figure 5B*, *Figure 5—figure supplement 1B*). To test whether this cycling D cell phenotype resulted from the presence of the DHB transgene or environmental stressors, such as temperature fluctuation, we examined the VPC lineage in animals lacking the CDK sensor at 25℃ and 28℃. At 25℃, we observed a low penetrance (2%) of cycling D cells in a strain expressing an endogenously tagged DNA licensing factor, CDT-1::GFP (*Figure 5B*, *Figure 5—figure supplement 1B*), which is cytosolic in cycling cells (*Matus et al., 2014*; *Matus et al., 2015*). From lineage analysis, L2 larvae, expressing a seam cell reporter (*scm*>GFP), that were temperature shifted from 20℃ to 28℃ displayed approximately a 30% occurrence of extra D cell divisions (*Figure 5—figure supplement 1C–E*). Lastly, we wanted to determine whether D cells that show $CDK^{inc}$ activity divide. To accomplish this, we collected time-lapses of DHB::GFP animals grown at 25℃. These time-lapses revealed 10 occurrences of D cells born into a $CDK^{inc}$ rather than a $CDK^{low}$ state (*Figure 5C and D*; *Figure 5—video 1*). In all 10 cases, the $CDK^{inc}$ D cell goes on to divide (*Figure 5—figure supplement 1A*). Thus, we find that CDK activity shortly after mitosis is a predictor of future cell behavior, even in rare stochastic cases of extra cell divisions in *C. elegans*, an organism with a well-defined cell lineage.

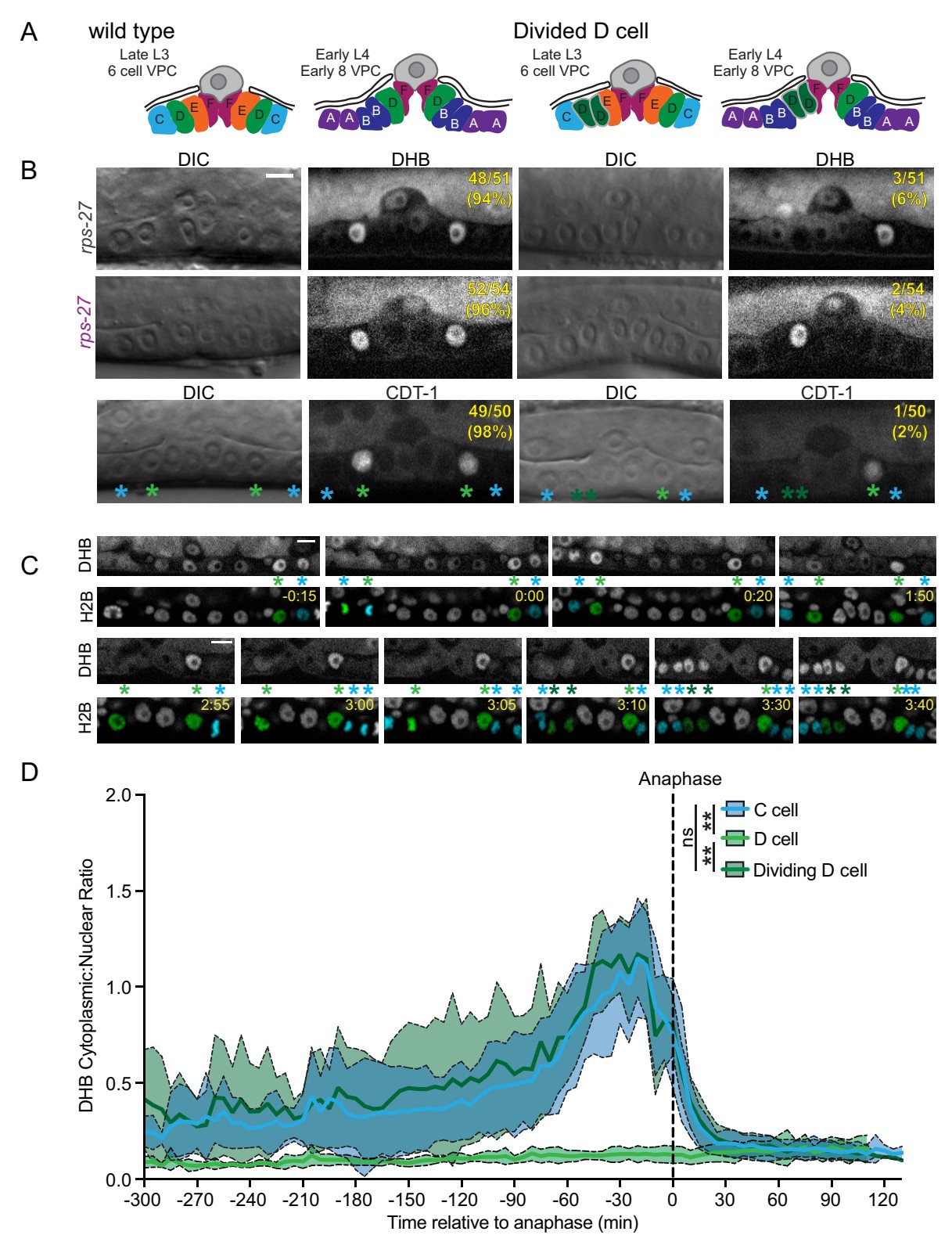

**Figure 5.** CDK activity predicts a cryptic stochastic fate decision in an invariant cell lineage. (**A**) Schematic of wild type vulva and vulva with a divided D cell. (**B**) Representative images at the Pn.p 6 cell stage from CDK sensor strains (top, middle) and endogenous *cdt-1::GFP* (bottom), showing wild type vulva on the left and vulva with a divided D cell on the right. Penetrance of each phenotype for each strain is annotated on the DHB image. (**C**) Frames from a time-lapse with a dividing D cell (left; see *Figure 3—video 1*). Nuclei (H2B) are highlighted in green for the D cell and cyan for the C cells.

*Figure 5 continued on next page*

**Figure 5 continued**

Green asterisks mark the D cell and cyan asterisks mark the C cell. Scale bar = 10 μm. (D) DHB ratio for C cell, quiescent D cell and dividing D cell (*n* = 10 quiescent D cells, *n* = 20 C cell divisions and *n* = 10 D cell divisions). Dotted line indicates time of anaphase. Line and shaded error bands depict mean ± SD. ns, not significant, **p≤0.01. Significance determined by statistical simulations; *p*-values in *Supplementary file 1*. The online version of this article includes the following video, source data, and figure supplement(s) for figure 5:

**Source data 1.** Source data for *Figure 5*.
**Figure supplement 1.** The vulval D cell divides stochastically.
**Figure 5—video 1.** Representative time-lapse of vulval D cell division expressing rps-27>DHB::GFP, Related to *Figure 5*, *Figure 5—figure supplement 1*.
https://elifesciences.org/articles/63265#fig5video1

## Generation of inducible CDK sensor transgenic lines in zebrafish

To investigate the predictive capability of DHB ratios in zebrafish, we generated two CDK sensor lines with different fluorescent protein combinations, DHB-mNeonGreen (DHB-mNG) and DHB-mScarlet (DHB-mSc) with H2B-mSc and H2B-miRFP670, respectively, to allow for flexibility with imaging and experimental design (*Figure 6A*). Both transgenes are under the control of the *hsp70l* heat shock-inducible promoter, which produces robust ubiquitous expression after shifting the temperature from 28.5 to 40℃ for 30 min (*Figure 6B*; *Halloran et al., 2000*; *Shoji et al., 1998*). We also generated a transgenic line, *Tg(ubb:Lck-mNG)*, that ubiquitously labels the plasma membrane with mNG, which we crossed into the HS:DHB-mSc-2A-H2B-miRFP670 line to simultaneously visualize CDK activity (DHB-mSc), segment nuclei (H2B-miRFP670) and segment the plasma membrane (LCK-mNG) (*Figure 6A*).

To verify that DHB localizes in a cell-cycle-dependent manner in both CDK sensor lines, we first used time-lapse microscopy and quantified DHB ratios across cell divisions in the tailbud of bud or 22 somite-stage embryos (*Figure 6C and D*, *Figure 6—figure supplement 1A*). We observed the expected localization pattern for both CDK sensor lines, with maximal nuclear exclusion of the sensor shortly before mitosis in G2 (3.42 ± 0.56 (DHB-mNG) and 6.57 ± 2.00 (DHB-mSc)) and low ratios (0.69 ± 0.17 (DHB-mNG) and 0.51 ± 0.21 (DHB-mSC)) representing nuclear accumulation of the sensor shortly after mitosis in G1 (*Figure 6F*, *Figure 6—figure supplement 1A*). To establish the DHB ratio for S phase we visualized PCNA-GFP in the tailbud of DHB-mSC embryos as PCNA forms puncta in the nucleus at S phase entry and returns to a uniform nuclear distribution in G2 (*Figure 6E*; *Leonhardt et al., 2000*; *Leung et al., 2011*). Approximately 38.5 min after puncta formation, corresponding to mid-S phase, the DHB ratio is 1.36 ± 0.36, which is significantly higher than the G1 DHB value (0.51 ± 0.21; *Figure 6F*). Thus, we conclude that both CDK sensor lines localize in a cell-cycle-dependent fashion, and that quantitative measurements can be used to determine interphase states.

Next, using both DHB transgenic lines, we examined CDK activity in a number of defined embryonic tissues. Imaging of the developing tailbud revealed cells in all phases of the cell cycle with mean DHB ratios of 1.95 ± 1.74 (mNG) and 1.67 ± 2.05 (mSC) (*Figure 7A and B*, *Figure 7—figure supplement 1A*). The tailbud of vertebrate embryos contain neuromesodermal progenitors (NMPs) (*Martin, 2016*), which in zebrafish have been reported to be predominantly arrested in the G2 phase of the cell cycle (*Bouldin et al., 2014*). Consistent with this, we observed cells with high CDK activity in the tailbud (orange arrows; *Figure 7A*, *Figure 7—figure supplement 1A*). This enrichment is eliminated when embryos are treated with the CDK4/6 inhibitor palbociclib, leading to a significant increase of cells in the tailbud with low CDK activity (0.58 ± 0.3), similar in range to the G1/G0 values we measured during time-lapse (0.69 ± 0.17; *Figure 7C–D*). We also made the surprising observation that primitive red blood cells in the intermediate cell mass of 24 hr post-fertilization (hpf) embryos, which are nucleated in zebrafish, display high CDK activity (3.00 ± 0.97) indicating that they are likely in the G2 phase of the cell cycle (*Figure 7—figure supplement 1E and F*), suggesting that cell-cycle regulation may be important for hematopoiesis (*Brönnimann et al., 2018*; *De La Garza et al., 2019*).

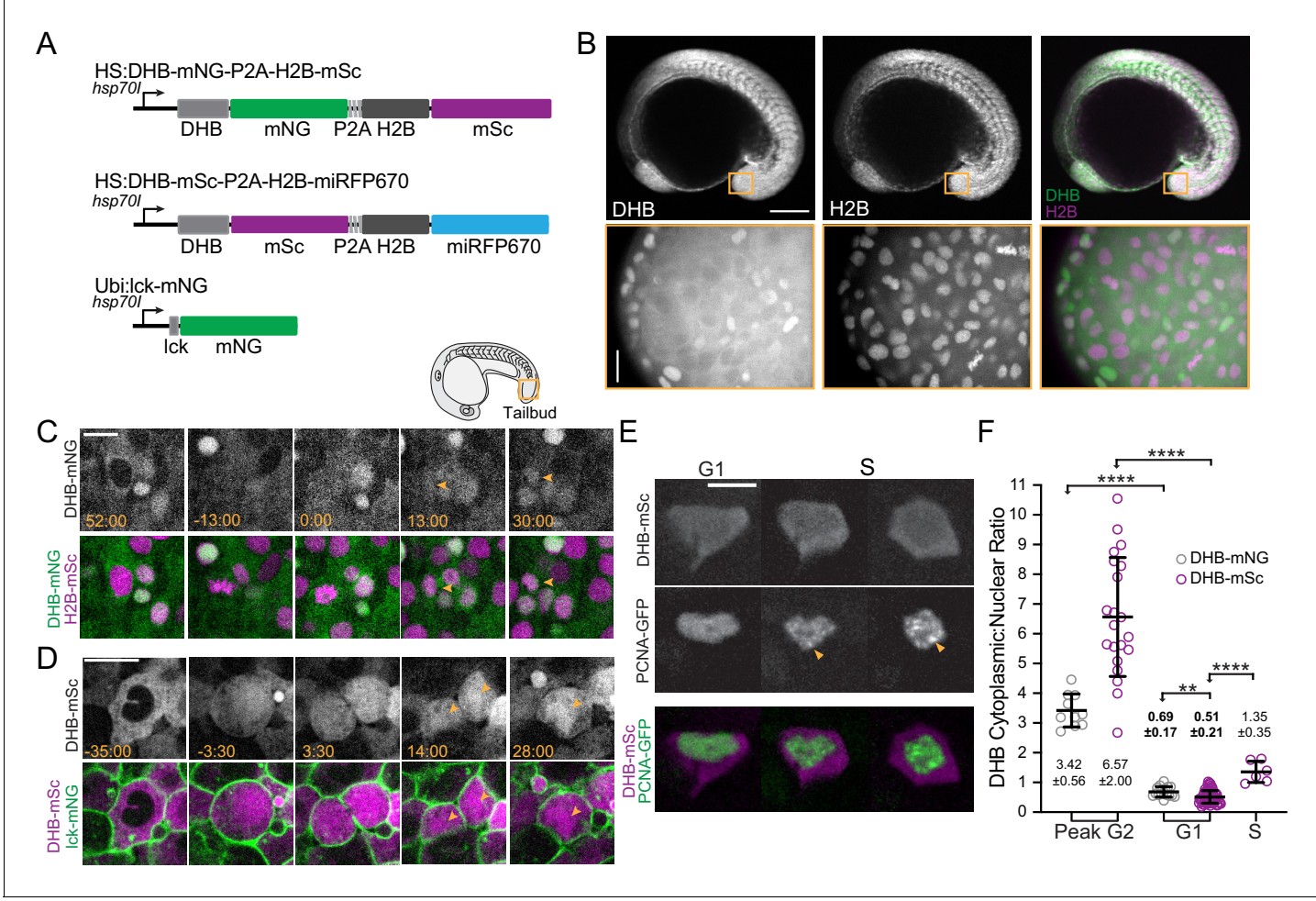

**Figure 6.** Generation of inducible CDK sensor transgenic lines in the zebrafish. (**A**) Schematics of inducible zebrafish variants of the CDK sensor fused to mNG (top) or mSc (middle) and a nuclear mask (H2B-FP) separated by a self-cleaving peptide (P2A). Schematic of inducible membrane marker (lck-mNG; bottom). (**B**) Representative images of HS:DHB-mNG-P2A-H2B-mSC at 18 somites. Scale bar top row = 250 μm. (**C, D**) Frames of DHB time-lapses taken from the developing tailbud as designated by the orange box shown in the schematic. (**E**) DHB-mSC and PCNA-GFP puncta during S phase. (**F**) Dot plot of DHB ratios during interphase states ($n \geq 7$ cells from $\geq 2$ embryos). Insets, orange box, are zoom-ins. Scale bar = 20 μm. Line and error bars depict mean ± SD. Numbers in bold are tissues in G0. ns, not significant, **$p \leq 0.01$, ****$p \leq 0.0001$. Significance determined by statistical simulations; *p*-values in *Supplementary file 1*.

The online version of this article includes the following video, source data, and figure supplement(s) for figure 6:

**Source data 1.** Source data for *Figure 6*.
**Source data 2.** Source date for *Figure 6—figure supplement 1*.
**Figure supplement 1.** CDK sensor expression in zebrafish developing tailbud.
**Figure 6—video 1.** Representative time-lapse of birth of CDK^low cells in the posterior growth zone of 22 somite-stage zebrafish, Related to *Figure 6*.
https://elifesciences.org/articles/63265#fig6video1

## Visualization of proliferation and quiescence during zebrafish development

To examine differences between proliferating and quiescent cells, we examined CDK activity in the somites, which are segmental mesodermal structures that give rise to skeletal muscle cells and other cell types (*Martin, 2016*), and adaxial cells, cells positioned at the medial edge of the somite next to the axial mesoderm (*Figure 7F and G*). The adaxial cells are the slow muscle precursors and are considered to be in a quiescent state through the cooperative action of Cdkn1ca (p57) and MyoD (*Osborn et al., 2011*). In the most recently formed somites at 24 hpf, cells can be observed in all

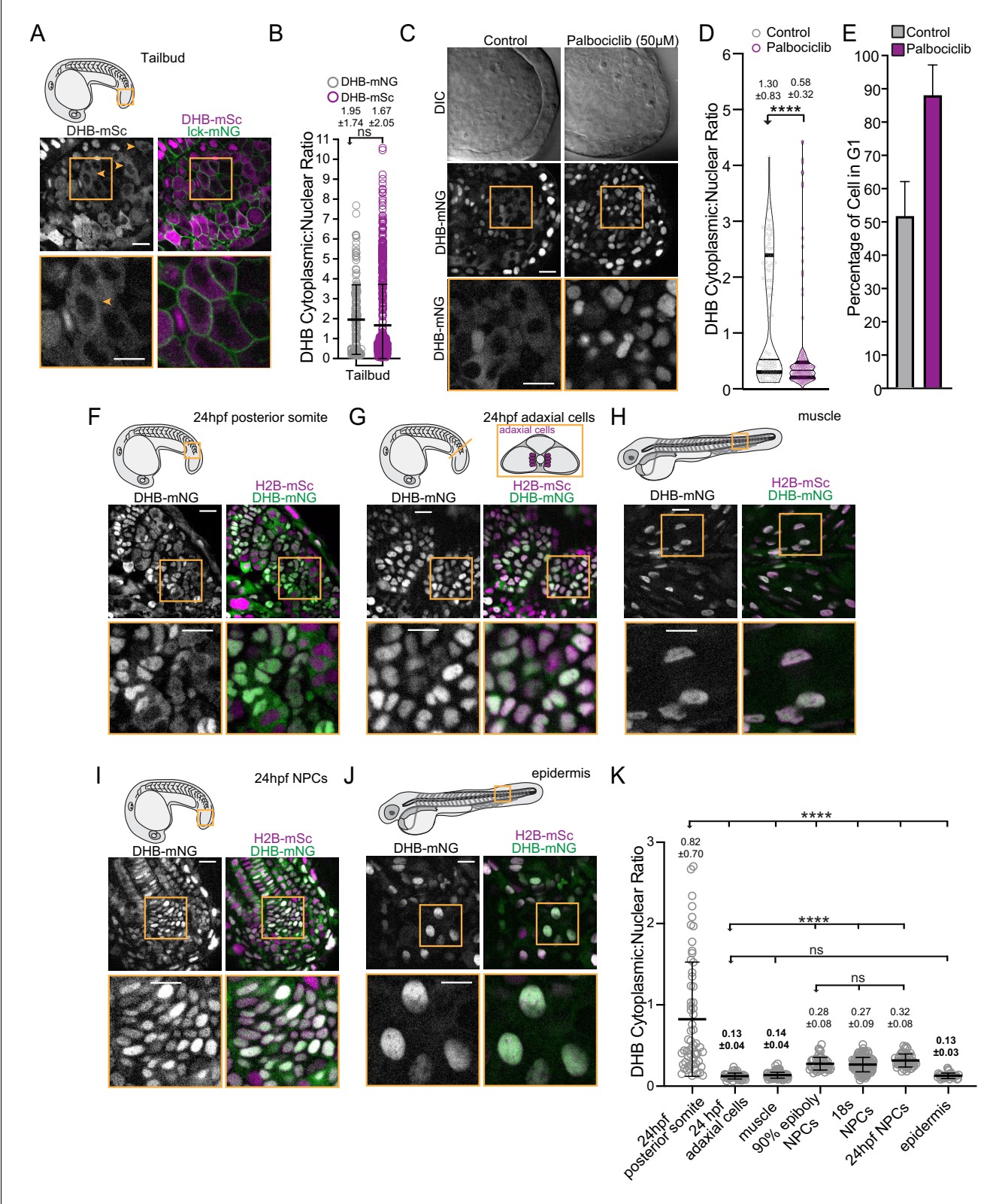

**Figure 7.** Visualization of CDK activity during zebrafish development. (**A**) Representative micrographs of CDK sensor (orange arrows and box inset highlights cytosolic CDK sensor localization) and quantification of DHB ratio (**B**) in the tailbud (*n* ≥ 160 cells). (**C**) Representative images of the tailbud of control or 50 μM palbociclib treated embryos (*n* ≥ 3 embryos). (**D**) Quantification of DHB in the tailbud (posterior wall and notochord cells excluded) of control or 50 μM palbociclib treated embryos at 20–22 somite stage. (**E**) Percentage of cells in G1 in the tailbud (posterior wall and notochord cells

*Figure 7 continued on next page*

*Figure 7 continued*

excluded) of control or 50 µM palbociclib treated embryos. (F–J) Representative micrographs of cells of 24 hpf posterior somites (F; $n \geq 59$ cells), adaxial cells (G; $n \geq 50$ cells), differentiated muscle at 72 hpf (H; $n \geq 101$ cells), notochord progenitors (NPCs) (I; $n \geq 48$ cells), and epidermis at 72 hpf (J; $n \geq 32$ cells). Insets, orange box, are zoom-ins. Scale bar = 20 µm. (K) Quantification of DHB ratios in zebrafish tissues. Line and error bars depict mean ± SD. ns, not significant, ****$p \leq 0.0001$. Orange boxes in schematics (A, F, G and H) depict region shown by the corresponding micrographs in each representative panel. In panel G, a schematic of a transverse section illustrating the position of adaxial cells is shown, but the micrograph is a lateral view. Significance determined by statistical simulations; *p*-values in ***Supplementary file 1***.

The online version of this article includes the following source data and figure supplement(s) for figure 7:

**Source data 1.** Source data for *Figure 7*.
**Source data 2.** Source date for *Figure 7—figure supplement 1*.
**Figure supplement 1.** CDK sensor expression in zebrafish and CDK4/6 inhibition in developing zebrafish increase percentage of cells in G1.

phases of the cell cycle (***Figure 7F and K***). Consistent with what we observed in the tailbud, treatment with palbociclib also caused somite cells to arrest with low CDK activity in G1/G0 (0.33 ± 0.42; ***Figure 7—figure supplement 1B–D***). As opposed to the majority of cells in the lateral regions of recently formed somites, adaxial cells possess low CDK activity (0.13 ± 0.04; ***Figure 7K***). At later stages, the majority of cells in the lateral regions of the somite will differentiate into fast skeletal muscles fibers, which are also considered to be in a quiescent state (***Halevy et al., 1995***). Examination of DHB ratios in 72 hpf skeletal muscle fibers revealed they have low CDK activity (0.14 ± 0.04 (mNG) and 0.13 ± 0.04 (mSc)), similar to the adaxial cells, but significantly different than the mean DHB ratios of undifferentiated cells at 24 hr (0.82 ± 0.70 (mNG) and 0.99 ± 0.084 (mSc); ***Figure 7H and K***, ***Figure 7—figure supplement 1G–I***). Thus, from our static imaging, we can identify cell types with low CDK activity that are thought to be quiescent.

We next sought to determine if we can differentiate between the G1 and G0 state based on ratiometric quantification of DHB. We compared adaxial cells to notochord progenitor cells, which are held transiently in G1/G0 before re-entering the cell cycle upon joining the notochord (***Figure 7I***; ***Sugiyama et al., 2014***; ***Sugiyama et al., 2009***). Notably, the mean DHB-mNG ratio of the notochord progenitors (0.32 ± 0.08) is significantly higher than the DHB-mNG ratio of the quiescent adaxial cells (0.13 ± 0.04; ***Figure 7F and J***). This elevated DHB ratio is consistent in notochord progenitors at two other earlier developmental stages, 90% epiboly (0.28 ± 0.08) and 18 somites (0.27 ± 0.09; ***Figure 7K***). The mean DHB-mSc ratio in the notochord progenitors (0.33 ± 0.08) is also significantly different than the differentiated epidermis (0.13 ± 0.03; ***Figure 7J***, ***Figure 7—figure supplement 1K–L***). Based on this difference in DHB ratios between notochord progenitors and differentiated cell types, including muscle and epidermis (***Figure 7K***), and our knowledge of the normal biology of these cells, we conclude that the CDK sensor can infer cell cycle state in the zebrafish, as it can distinguish between a cycling G1 state and a quiescent G0 state.

## A bifurcation in CDK activity at mitotic exit is conserved in *C. elegans* and zebrafish

We next investigated whether zebrafish cells separate into G1/CDK$^{inc}$ and G0/CDK$^{low}$ populations as they do in the nematode *C. elegans* and whether these CDK-activity states are a general predictor of future cell behavior in both animals. First, we plotted all of the time-lapse CDK sensor data we collected in *C. elegans* (***Figure 8A and B***) and zebrafish (***Figure 8C***). For *C. elegans*, plotting of all CDK sensor trace data, irrespective of lineage, demonstrated that cells entering a CDK$^{low}$ state after mitosis corresponded to quiescent cells, while cells that exited mitosis into a CDK$^{inc}$ state corresponded to cells from pre-terminal divisions. For zebrafish, in a lineage agnostic manner, we plotted all the traces from the tailbud. We classified cells as CDK$^{low}$ that remained below 0.19, the upper bound of the DHB ratio for quiescent adaxial cells at this stage of development (***Figure 7K***), for three or more frames post-anaphase. Indeed, we found that these traces could also be classified into CDK$^{low}$ and CDK$^{inc}$ populations (***Figure 8C***). In addition to a fast cycling population of cells at this developmental stage, we also identified cells that maintain a CDK$^{inc}$ DHB ratio but appear to stay in a prolonged G1 phase, potentially representing a slow cycling population of cells.

As we were able to detect a rare stochastic lineage change in the *C. elegans* vulval lineage (***Figure 5***), we selected all CDK sensor trace data from the *C. elegans* VPCs (***Figure 8—figure supplement 1A***) and used this data to build a classifier to predict proliferative (G1) versus quiescent (G0)

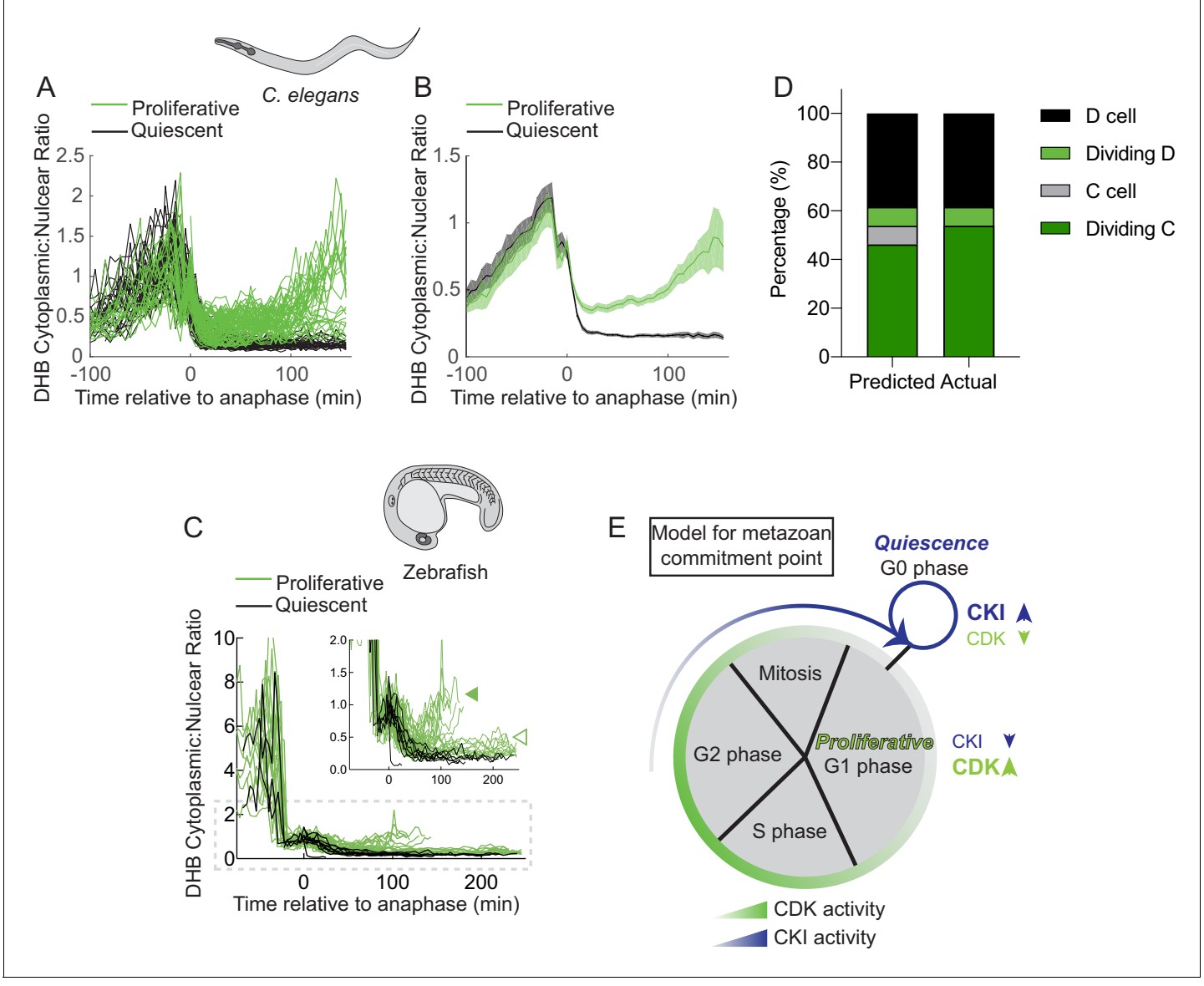

**Figure 8.** A bifurcation in CDK activity at mitotic exit predicts the proliferation-quiescence decision. (A–D) Single-cell traces of CDK activity for all quantified *C. elegans* (A, B) and zebrafish (C) cell births for CDK^inc cells (green) and CDK^low cells (black). DHB ratio of single-cell data (A, C) and mean ± 95% confidence intervals (B) are plotted for each cell analyzed relative to anaphase. A solid green arrowhead indicates a population of fast cycling CDK^inc cells while the open green arrowhead indicates a population of CDK^inc cells that may be slow cycling in an extended G1 phase. (E) A stacked bar graph of predicted vs. actual cell fates for the D, dividing D, C, and dividing C cells, based on a classifier trained on post-anaphase CDK activity in VPC trace data. (E) A model for the metazoan commitment point argues that the G1/G0 decision is influenced by a maternal input of CKI activity and that CDK activity shortly after mitotic exit can determine future cell fate.

The online version of this article includes the following figure supplement(s) for figure 8:

**Figure supplement 1.** Statistical evaluation of the predictive model for future cell behavior in *C. elegans* and a schematic describing the method used for statistical simulations.

cell fates based on CDK activity after anaphase (*Figure 8—figure supplement 1A–C*). Cross-examining our modeling with the known VPC lineage demonstrated that at 20 min after anaphase we had 85% accuracy in predictions with near-perfect prediction 60 min after anaphase (*Figure 8—figure supplement 1B*). To test the predictive power of the classifier, we analyzed CDK trace data from the births of C and D cells, where some D cells stochastically divide (*Figure 5*). Our classifier correctly predicted cell fate 92% ($n$ = 24/26 single-cell traces) of the time, including the two occurrences of a

stochastic mitotic D cell in the data set (*Figure 8D*). Finally, to determine whether we could predict future cell behavior independent of cell type, we trained a new classifier using 75% of all collected *C. elegans* time-lapse trace data from the SM, uterine, and VPC lineages. We used the remaining 25% of traces as test data. When cross-referenced with the known *C. elegans* lineage, our cell-type agnostic classifier correctly predicted the difference between a CDK$^{inc}$ proliferative cell and a CDK$^{low}$ quiescent cell 93% (62/67) of the time.

Together, these results demonstrate that during development, cycling cells encounter a bifurcation in CDK activity following mitosis where they either: (1) increase in CDK activity and become poised to cycle, or (2) exit into a CDK$^{low}$ state and undergo cell-cycle quiescence (*Figure 8E*). Thus, we suggest a model where cells from developing tissue in *C. elegans* and zebrafish must cross an early commitment point in the cell cycle where these cells must make the decision to divide or enter G0. The decision to undergo quiescence is crucial to tissue integration and organization and is in part likely controlled by the activity of evolutionarily conserved CKI(s) in the mother cell that control daughter cell CDK activity (*Figure 8E*).

## Discussion

### A CDK sensor for live-cell in vivo imaging of interphase states and the G1/G0 Transition

We introduce here a CDK-activity sensor to visually monitor interphase and the proliferation-quiescence decision in real-time and in vivo in two widely used research organisms, *C. elegans* and zebrafish. This sensor, which reads out the phosphorylation of a DHB peptide by CDKs (*Hahn et al., 2009*; *Spencer et al., 2013*), allows for quantitative assessment of cell cycle state, including G0. The use of FUCCI in zebrafish (*Bouldin and Kimelman, 2014*; *Sugiyama et al., 2009*) and past iterations of a CDK sensor in *C. elegans* (*Deng et al., 2020*; *Dwivedi et al., 2019*; *van Rijnberk et al., 2017*) and *Drosophila* (*Hur et al., 2020*) have been informative in improving our understanding of cell-cycle regulation of development, but have not addressed the proliferation-quiescence decision. The DHB transgenic lines generated in this study will allow researchers to distinguish G1 from G0 shortly after a cell has divided and directly study G0-related cell behaviors, such as quiescence, terminal differentiation, and senescence, in living organisms.

Previously, CDK sensors have been used to distinguish between proliferative and quiescent cells in asynchronous mammalian cell culture populations (*Arora et al., 2017*; *Cappell et al., 2016*; *Gast et al., 2018*; *Gookin et al., 2017*; *Miller et al., 2018*; *Moser et al., 2018*; *Overton et al., 2014*; *Spencer et al., 2013*; *Yang et al., 2015*). As mammalian cells complete mitosis, they are born into either a CDK2$^{inc}$ state in which they are more likely to divide again or a CDK2$^{low}$ state in which they remain quiescent. Here we have examined the CDK activity state of cells in an invertebrate with a well-defined and invariant lineage, *C. elegans*, and a vertebrate that lacks a defined cell lineage, the zebrafish. In both contexts, we can visually and quantitatively differentiate between cells that are in a CDK$^{inc}$ state following cell division and cells that are in a CDK$^{low}$ state. Strikingly, in *C. elegans* these states precisely correlate with the lineage pattern of the three post-embryonic tissues we examined: the SM cells, uterine cells, and VPCs. Cells born into a CDK$^{inc}$ state represented pre-terminal divisions, whereas cells born into a CDK$^{low}$ state were quiescent and represented cells that had undergone their terminal division. By distinguishing these two states in CDK activity, we were able to accurately identify shortly after cell birth a rare stochasticity that was first described through careful end-point lineage analysis nearly 36 years ago in the *C. elegans* vulval lineage (*Sternberg, 1984*; *Sternberg and Horvitz, 1986*). Further, statistical modeling demonstrated that we could predict future cell behavior with >85% accuracy in *C. elegans* just 20 min post-anaphase. From static imaging in zebrafish, we found that we could readily distinguish between CDK$^{inc}$ cells in G1, such as notochord progenitors, which re-enter the cell cycle after joining the notochord, and quiescent tissues that contain CDK$^{low}$ cells in G0, such as skeletal muscle and epidermis. While analysis of time-lapse data did lead to the identification of cells born into either CDK$^{inc}$ or CDK$^{low}$ states, we were unable to follow and quantify enough cell births to determine whether CDK activity at mitotic exit is also predictive of future proliferation behavior during zebrafish development. We attribute our inability to capture an adequate number of cell births largely to a combination of conventional confocal microscopy and manual cell tracking and quantification. Nonetheless, in both organisms the

CDK sensor can be easily used to separate G1 from G0 without the need for multiple fluorescent reporters (*Bajar et al., 2016*; *Oki et al., 2015*) or fixation followed by antibody staining for FACS analysis (*Tomura et al., 2013*).

## In vivo evidence of a G2 commitment point in the metazoan cell cycle

The classic model of the Restriction Point, the point in G1 at which cells in culture decide to commit to the cell cycle and no longer require growth factors (e.g. mitogens), is that mammalian cells are born uncommitted and that the cell-cycle progression decision is not made until several hours after mitosis (*Jones and Kazlauskas, 2001*; *Pardee, 1974*; *Zetterberg and Larsson, 1985*; *Zwang et al., 2011*). An alternative model has been proposed in studies using single-cell measurements of CDK2 activity in asynchronous populations of MCF10A cells (*Spencer et al., 2013*) and other nontumorigenic as well as tumorigenic cell lines (*Moser et al., 2018*). This model extends the classic Restriction Point model for cell cycle commitment. During the G2 phase of the cell cycle, the mother cell is influenced by levels of p21 and cyclin D and these levels affect the phosphorylation status of Rb in CDK$^{low}$ and CDK$^{inc}$ daughter cells, respectively (*Min et al., 2020*; *Moser et al., 2018*). In CDK$^{low}$ daughter cells, phospho-Rb is low, and these cells are still sensitive to mitogens. Whether cells in vivo coordinate cell-cycle commitment with levels of CKI and CDK over this extended Restriction Point was poorly understood.

By first quantifying the cytoplasmic:nuclear ratio of the CDK sensor in time-lapse recordings of cell divisions in *C. elegans* somatic lineages, we were able to use DHB ratios as a proxy for CDK levels to distinguish two populations of daughter cells: the first being actively cycling cells in a CDK$^{inc}$ state (G1) and the second being quiescent cells in a CDK$^{low}$ state (G0). We then quantified cytoplasmic:nuclear ratio of the CDK sensor in time-lapse recordings of cell divisions in zebrafish and we were also able to distinguish two populations of daughter cells. As data from asynchronous cell culture studies suggest that the decision to commit to the cell cycle is made by the mother cell as early as G2 (*Moser et al., 2018*; *Spencer et al., 2013*), we wanted to determine if this same phenomenon occurred in vivo. To accomplish this, we endogenously tagged one of two CKIs in the *C. elegans* genome, *cki-1*, with GFP using CRISPR/Cas9. We paired static live-cell imaging of GFP::CKI-1 with DHB::2xmKate2 during vulval development. Similar to in vitro experiments (*Moser et al., 2018*; *Spencer et al., 2013*), we found that mother cells whose daughters are born into a CDK$^{inc}$ G1 state will divide again, expressing low levels of GFP::CKI-1. In contrast, mother cells of daughters that will exit the cell cycle express a peak of GFP::CKI-1 in G2 which increases as daughter cells are born into a CDK$^{low}$ G0 state. Thus, our data demonstrate that an extended Restriction Point exists in the cell cycle of intact Metazoa. Furthermore, the in vivo proliferation-quiescence decision can be predicted in *C. elegans* by CDK activity shortly after mitotic exit and, based on our gain-of-function studies, is highly sensitive to levels of CKI-1 shortly before and after the mother cell divides.

## Conclusion

We demonstrate here that the CDK sensor functions in both *C. elegans* and zebrafish to read out cell cycle state dynamically, and unlike other in vivo cell-cycle sensors, can distinguish between proliferative and quiescent cells within an hour of cell birth. As nematodes and vertebrates last shared a common ancestor over 500 million years ago, this suggests that the CDK sensor is likely to function in a similar fashion across Metazoa. With advances in time-lapse in vivo 4D imaging and machine learning methods that facilitate the collection and analyses of CDK sensor activity in 4D, we envision an increased demand for this tool to study cell-cycle-regulated biology in other animals. The broad functionality of the sensor will offer researchers a unique opportunity to dissect the relationship between cell cycle state and cell fate during normal development, cellular reprogramming, and tissue regeneration. Finally, as an increasing body of evidence suggests that cell cycle state impinges on morphogenetic events ranging from gastrulation (*Grosshans and Wieschaus, 2000*; *Murakami et al., 2004*), convergent extension (*Leise and Mueller, 2004*) and cellular invasion (*Kohrman and Matus, 2017*; *Matus et al., 2015*; *Medwig-Kinney et al., 2020*), this CDK sensor will provide the means to increase our understanding of the relationship between interphase states and morphogenesis during normal development and diseases arising from cell-cycle defects, such as cancer.

## Supplemental materials

This manuscript is accompanied by *Supplementary file 1*, a spreadsheet of all reported *p*-values from statistical tests performed.

# Materials and methods

**Key resources table**

| Reagent type (species) or resource | Designation | Source or reference | Identifiers | Additional information |
|---|---|---|---|---|
| Strain, strain background (*E. coli*) | *cdk-1* RNAi | *Rual et al., 2004* | | |
| Strain, strain background (*C. elegans*) | DQM298 | This study | | LoxN::rps-27> DHB::GFP::P2A:: H2B::2xmKate2 (bmd86) LGI |
| Strain, strain background (*C. elegans*) | DQM394 | This study | | LoxN::rps-0>DHB::mKate2 (bmd118) LGII |
| Strain, strain background (*C. elegans*) | DQM406 | This study | | LoxN::hsp16—41> cki-1::2xmTagBFP2 (bmd129) LGI; LoxN::rps-0>DHB::mKate2 (bmd118) LGII |
| Strain, strain background (*C. elegans*) | DQM543 | This study | | LoxN::rps-27> DHB::2xmKate2::P2A::H2B::GFP (bmd147) LGI |
| Strain, strain background (*C. elegans*) | DQM662 | This study | | LoxN::pcn-1>PCN-1::GFP (bmd200) LGI; LoxN::rps-27> DHB::2xmKate2 (bmd168) LGII |
| Strain, strain background (*C. elegans*) | DQM586 | This study | | LoxN::rps-27> DHB::2xmKate2 (bmd156) LGI; GFP::LoxN::CKI-1::3xFLAG (bmd132) LGII |
| Strain, strain background (*C. elegans*) | JLF634 | This study | | cdt-1::ZF::LoxP::GFP::3xFLAG (wow98) LGI; zif-1(gk117) LG III |
| Strain, strain background (*C. elegans*) | JR667 | CGC | | SCM> GFP (wIs51) LGV |
| Strain, strain background (*D. rerio*) | Sbu108 | This study | | Tg(hsp70l:DHB.mNeonGreen-p2a-H2B.mScarlet) |
| Strain, strain background (*D. rerio*) | Sbu107 | This study | | Tg(ubb:Lck.mNeonGreen) |
| Strain, strain background (*D. rerio*) | Sbu109 | This study | | Tg(hsp70l:DHB.mScarlet-p2a-H2B.miRFP670) |
| Recombinant DNA reagent | Plasmid: pAWK61 | This study | RRID:Addgene_163642 | NotI-rps-27> DHB-ClaI-GFP-P2A-H2B-2xmKate2-NheI-3xHA (I) |
| Recombinant DNA reagent | Plasmid: pAWK41 | This study | | NotI-rps-0>DHB-mKate2(GLO)-NheI-3xHA (II) |
| Recombinant DNA reagent | Plasmid: pWZ186 | This study | RRID:Addgene_163641 | NotI-rps-27> DHB-2xmKate2-P2A-H2B-GFP-NheI-3xHA (I) |
| Recombinant DNA reagent | Plasmid: pWZ194 | This study | RRID:Addgene_163640 | NotI-rps-27> DHB-2xmKate2 (I) |
| Recombinant DNA reagent | Plasmid: pTNM054 | This study | | NotI-rps-27> DHB-2xmKate2 (II) |
| Recombinant DNA reagent | Plasmid: pWZ111 | This study | | NotI-ccdB-ClaI-GFP-3xHA (I) |

*Continued on next page*

*Continued*

| Reagent type (species) or resource | Designation | Source or reference | Identifiers | Additional information |
|---|---|---|---|---|
| Recombinant DNA reagent | Plasmid: pWZ157 | This study | RRID:Addgene_163639 | *NotI-pcn-1>PCN-1-GFP-3xHA (I)* |
| Recombinant DNA reagent | Plasmid: pWZ123 | This study | | *hsp-16.41-NotI-ccdB-ClaI-2xmTagBFP2-NheI-3xHA (I)* |
| Recombinant DNA reagent | Plasmid: pWZ199 | This study | | *hsp-16.41> CKI-1-2xmTagBFP2-NheI-3xHA (I)* |
| Recombinant DNA reagent | Plasmid: pDD282 | *Dickinson et al., 2015* | RRID:Addgene_66823 | *ccdB-GFP-C1SEC3xFLAG- ccdB* |
| Recombinant DNA reagent | Plasmid: pNJP026 | This study | | *GFP-C1SEC3xFLAG-cki-1 (II)* |
| Recombinant DNA reagent | Plasmid: pWZ143 | This study | | *cki-1 sgRNA* |
| Recombinant DNA reagent | Plasmid: pJF250 | *Sallee et al., 2018* | | *ccdB-ZF-GFP-SEC-3xFLAG-ccdB* |
| Recombinant DNA reagent | Plasmid: pMS254 | This study | | *cdt-1-ZF-GFP-3xFLAG (I)* |
| Recombinant DNA reagent | Plasmid: pMS250 | This study | | *cdt-1 sgRNA* |
| Recombinant DNA reagent | Plasmid: pRM14 | This study | RRID:Addgene_163693 | *hsp70I-DHB-mNeonGreen-P2A-h2b::mScarlet* |
| Recombinant DNA reagent | Plasmid: pRM27 | This study | RRID:Addgene_163695 | *ubb-Lck-mNeonGreen* |
| Recombinant DNA reagent | Plasmid: pRM15 | This study | RRID:Addgene_163694 | *hsp70I-DHB-mScarlet-P2A-h2b::miRFP670* |
| Recombinant DNA reagent | Plasmid: HS-PCNA-GFP | *Strzyz et al., 2015* | RRID:Addgene_105942 | *HS-PCNA-GFP* |
| Sequenced-based reagent | *cki-1* sgRNA | This study | | gaagacatttgaaaagagtg |
| Sequenced-based reagent | *cdt-1* sgRNA | This study | | ggatggccgtggtgtgtgg |
| Sequenced-based reagent | DQM205 | This study | | Primer: *rps-27* F to insert in NotI site of pAP88 catcctgtaaaacgacggccagtgc TTCAATCGGTTTTTCCTTG |
| Sequenced-based reagent | DQM206 | This study | | Primer: *rps-27* R to insert in NotI site of pAP88 ctcttttgacatacttcgggtagcggccgc TTTTATTCCACTTGTTGAGC |
| Sequenced-based reagent | DQM728 | This study | | Primer: *rps-0* F catcctgtaaaacgacggccagtgc GAGGAATGAAGAAATTTGC |
| Sequenced-based reagent | DQM729 | This study | | Primer: *rps-0* R cggaccaggtgacgtcgttggtcat ATTACCTTAAAATTCAAAAATTAATTTCAG |
| Sequenced-based reagent | DQM622 | This study | | Primer: *pcn-1* F to insert into NotI-ccdB-ClaI site of pWZ111 CATCCtgtaaaacgacggccagtgcGGCCGCagaaacagtggccgtattgg |
| Sequenced-based reagent | DQM609 | This study | | Primer *pcn-1* R to insert into NotI-ccdB-ClaI site of pWZ111 tgaacaattcttctcctttactcatcgatgctccGTCCATATTCTCGTCGTC |
| Sequenced-based reagent | DQM288 | This study | | Primer: *hsp* F catcctgtaaaacgacggccagtgc CACCAAAAACGGAACGTTGAGC |
| Sequenced-based reagent | DQM289 | This study | | Primer: *hsp* R ctcttttgacatacttcgggtagcggccg CCAATCCCGGGGATCCGA |
| Sequenced-based reagent | DQM303 | This study | | Primer: *cki-1* F atccccgggattggcggccgc ATGTCTTCTGCTCGTCGTTG |

*Continued on next page*

*Continued*

| Reagent type (species) or resource | Designation | Source or reference | Identifiers | Additional information |
|---|---|---|---|---|
| Sequenced-based reagent | DQM304 | This study | | Primer: *cki-1* R aatcaattccgaaaccattgaggctcccgatgctcc GTATGGAGAGCATGAAGATCG |
| Sequenced-based reagent | WZ1 | This study | | Primer: *gfp::cki-1* F atgttacccatccaactatacacc |
| Sequenced-based | NP63R | This study | | Primer: *gfp::cki-1* R gtggttctgacagtgagaac |
| Sequenced-based reagent | DQM490 | This study | | Primer: *cki-1* sgRNA tcctattgcgagatgtcttggaagacatttgaaaagagtg GTTTTAGAGCTAGAAATAGC |
| Sequenced-based reagent | oMS-219-F | This study | | Primer: *cdt-1* 5'HA (homology arm) F ttgtaaaacgacggccagtcg |
| Sequenced-based reagent | oMS-220-R | This study | | Primer: *cdt-1* 5'HA R CATCGATGCTCCTGAGGCTCC |
| Sequenced-based reagent | oMS-208-F | This study | | Primer: *cdt-1* 3'HA F CGTGATTACAAGGATGACGATGACAAGAGATAA aaactaatttctaagccatttgtaactaattttctcact |
| Sequenced-based reagent | oMS209-R | This study | | Primer: *cdt-1* 3'HA R ggaaacagctatgaccatgttatcga tttcccaacgaggcgattactgagc |
| Sequenced-based reagent | oMS-205-F | This study | | Primer: *cdt-1* sgRNA F GGATGGCCGTGGTGTGTGGgttttagagctagaaatagcaagt |
| Sequenced-based reagent | oJF436-R | This study | | Primer: *cdt-1* sgRNA R CAAGACATCTCGCAATAGG |
| Sequenced-based reagent | RM112 | This study | | Primer: *mNG-Lck* F ATGGGCTGCGTGTGCAGCAGCAACCCCGAGAT GGTGAGCAAGGGCGA |
| Sequenced-based reagent | RM113 | This study | | Primer: *mNG-Lck* R CTTGTACAGCTCGTCCATGC |
| Sequenced-based reagent | RM114 | This study | | Primer: *mNG-Lck* homology F ATCTTACTTTGAATTTGTTTACAGGgatccATGG GCTGCGTGTGCAGCAG |
| Sequenced-based reagent | RM115 | This study | | Primer: *mNG-Lck* homology R TCATGTCTGGATCATCATCGATCTTGTACAG CTCGTCCATGCCC |
| Sequenced-based reagent | RM169 | This study | | Primer: *DHB:mNG* F ATCTTACTTTGAATTTGTTTACAGGgatccatg acaaatgatgtcacctggagc |
| Sequenced-based reagent | RM173 | This study | | Primer: *DHB:mNG* R GGTGGCGACCGGTGGAAC |
| Sequenced-based reagent | RM174 | This study | | Primer: *mScarlet:CAAX* F GTTCCACCGGTCGCCACCATGGTGAGCAAGGGCGAG |
| Sequenced-based reagent | RM175 | This study | | Primer: *mScarlet:CAAX* R CTTATCATGTCTGGATCATCATCGATCTTGTACAGCT CGTCCATGCC |
| Sequenced-based reagent | RM192 | This study | | Primer: *DHB* F AAGCTACTTGTTCTTTTTGCAGGATCCATGACAAATGATG TCACCTGGAGCGAG |
| Sequenced-based reagent | RM193 | This study | | Primer: *DHB* R GCCGCTGCCCTGGGCC |
| Sequenced-based reagent | RM194 | This study | | Primer: *mScarlet* F GCCCAGGGCAGCGGCATGGTGAGCAAGGGCGAG |
| Sequenced-based reagent | RM195 | This study | | Primer: *mScarlet* R GTTGGTGGCGCCGCTGCCCTTGTACAGCTCGTCCATGCC |
| Sequenced-based reagent | RM196 | This study | | Primer: *P2A:H2B* F GGCAGCGGCGCCACC |

*Continued on next page*

*Continued*

| Reagent type (species) or resource | Designation | Source or reference | Identifiers | Additional information |
|---|---|---|---|---|
| Sequenced-based reagent | RM197 | This study | | Primer: *P2A:H2B* R GGTGGCGACCGGTGGAACCT |
| Sequenced-based reagent | RM198 | This study | | Primer: *miRFP670* F AGGtTCCACCGGTCGCCACCATGGTAGCAGGTCATGCCTC |
| Sequenced-based reagent | RM199 | This study | | Primer: *miRFP670* R CTTATCATGTCTGGATCATCATCGATTTAGCTCTCAAGCGCGGTGA |
| Sequenced-based reagent | *Codon-optimized DHB (with synthetic introns)* | This study | IDT | catcctgtaaaacgacggccagtgcggccgcATGACCAACGACGT CACCTGGTCCGAGGCCTCCTCCCCAGACGAGCGTACCCT CACCTTCGCCGAGCGTTGGCAACTCTCCTCCCCAGACGG AGTCGACACCGACGACGACCTCCCAAAGTCCCGTGCCTC CAAGCGTACCTGCGGGAGTCAACGACGACGAGTCCCCATC CAAGgtaagtttaaacatatatatactaactaaccctgattatttaaatt ttcagATCTTCATGGTCGGAGAGTCCCCACAAGTCTCCTCC CGTCTCCAAAACCTCCGTCTCAACAACCTCATCCCACGTC AACTCTTCAAGCCAACCGACAACCAAGAGACCGGAGCAT CGGGAGCCTCAGGAGCATCGATGAGTAAAGGAGAAGAAT TGTTCA |
| Sequenced-based reagent | *NgoMIV-P2A (codon-de-optimized)-his-58-GFP-NheI* | This study | Twist Biosciences | CATCCAAGCTCGGACATCGTGCCGGCGCGGGAAGTGGGGCCACG AACTTCAGTCTCCTCAAACAAGCCGGGGACGTCGAAGAGAACCCC GGGCCAATGCCACCAAAGCCATCTGCCAAGGGAGCCAAGAAGGC CGCCAAGACCGTCGTTGCCAAGCCAAAGGACGGAAAGAAGAGAC GTCATGCCCGCAAGGAATCGTACTCCGTCTACATCTACCGTGTTC TCAAGCAAGTTCACCCAGACACCGGAGTCTCCTCCAAGGCCATG TCTATCATGAACTCCTTCGTCAACGATGTATTCGAACGCATCGCT TCGGAAGCTTCCCGTCTTGCTCATTACAACAAACGCTCAACGAT CTCATCCCGCGAAATTCAAACCGCTGTCCGTTTGATTCTCCCAG GAGAACTTGCCAAGCACGCCGTGTCTGAGGGAACCAAGGCCGT CACCAAGTACACTTCCAGCAAGATGAGTAAAGGAGAAGAATTGT TCACTGGAGTTGTCCCAATCCTCGTCGAGCTCGACGGAGACGT CAACGGACACAAGTTCTCCGTCTCCGGAGAGGGAGAGGGAGA CGCCACCTACGGAAAGCTCACCCTCAAGTTCATCTGCACCACC GGAAAGCTCCCAGTCCCATGGCCAACCCTCGTCACCACCTTCT GCTACGGAGTCCAATGCTTCTCCCGTTACCCAGACCACATGAA GCGTCACGACTTCTTCAAGTCCGCCATGCCAGAGGGATACGTC CAAGAGCGTACCATCTTCTTTAAGgtaagtttaaacatatatatactaa ctactgattatttaaattttcagGACGACGGGAAACTACAAGACCCGTGC CGAGGTCAAGTTCGAGGGAGACACCCTCGTCAACCGTATCGAG CTCCAGgtaagtttaaacagttcggtactaactaaccatacatatttaaattttt cagGGAATCGACTTCAAGGGAGGACGGGAAACATCCTCGGACACA AGCTCGAGTACAACTACAACTCCCACAACGTCTACATCATGGCC GACAAGCAAAAGAACGGAATCAAGGTCAACTTCAAGgtaagttta aacatgatttttactaactaactaatctgatttaaattttcagATCCGTCACAAC ATCGAGGACGGATCCGTCCAACTCGCCGACCACTACCAACAAAA CACCCCAATCGGAGACGGACCAGTCCTCCTCCCAGACAACCACT ACCTCTCCACCCAATCCGCCCTCTCCAAGGACCCAAACGAGAAG CGTGACCACATGGTCCTCCTCGAGTTCGTCACCGCCGCCGGAAT CACCCACGGAATGGACGAGCTCTACAAGTCAGGAGCTAGCGGAG CCTACCCTTACGACG |

*Continued on next page*

*Continued*

| Reagent type (species) or resource | Designation | Source or reference | Identifiers | Additional information |
|---|---|---|---|---|
| Sequenced-based reagent | *gfp::cki-1 left homology arm* | This study | Twist Biosciences | ACGTTGTAAAACGACGGCCAGTCGCCGGCACTCACTGTCACCAAATGTACCGTATTGCTTTCCGGCTGTTATTGTTGTTATCACTGCTTCTTCTTCCTATCATGTTACCCATCCAACTATACACCTTAGACTAGTCATCTTATTGATATACATTCCTCCCATCCAACACAACGGTATTCTATTTATTTATCCAATTAGTCATAGTCGTACCACCATCCAGCACGAAGGTGCCTCTTTAGTAAAGAGTAGAAAGAAGAACCGGATGGGAAATGTTTTTGTTACAAAAATGACACATATTGTAGTGGACAGAAGGAGTGAGACAGACATGAGCAAGCCAATTTGTTTATAATTTCTCTTCTAGAAAAAAATACATTTTTCCATACTTCACTAGTCAAAACCTTTCACCTTTCTAATACATCTCGTAAACCATAATCTTGATAGTTCTGAGCATTTCAATACGAAAGCTTCTCACTGTCTAGATCTCTGACTGAGTGCCCTCATCAAAAGTGCAATCTGTCATCTGTTTCCTCATAATCACGGAGCACTAATTTTTCTCTCTGCGTCTCTATAATCAGATATCTCTCGTCACTAAGAACTTTCCGAAATGTTTATGCTTCTCATCTGACCACTTCGGTTCCGCACAAAAAAGTACGGCATTCCAAAAGAAATCTGATCCCCCTCCGTTCATTCGTGGTCCGAGTCGGTGCCACCAGTCGTTGCGCATTGAATATTTGTTTGGTCCGTTCCCCTTCTTCTCCGACTGCTGACCTCGGGCACTTTGATGACCGGGCCACCACCTCAGTACCCCTCTATTACACCCTCTTTGCCTCCGCGCATATGACTCCACCCCTTCTCGTGGAAGGCGTGTATCTCCCCTCTTTTCCGCTATTCCCTCGATGGATATATATTCAAATGTATGTGTGTTCCTGACGGGAGGGCGTCTCGCTTGAGAGCATCGTCACATCTTTTACAATTTTACTTATGATTTTACTTCATCTTCTTCTTCTTACTGCGATTTTGATATGCATTCTTATGTAAACTATTATTATTCCAGGTTTCCTCACTCTTTTCAAATGAGTAAAGGAGAAGAATTGTTCACTGGAG |
| Sequenced-based reagent | *gfp::cki-1 right homology arm* | This study | Twist Biosciences | GCGTGATTACAAGGATGACGATGACAAGAGAATGTCTTCTGCTCGTCGTTGCCTTTTCGGTCGTCCGACGCCCGAGCAACGCTCCAGGACTCGAATTTGGCTTGAAGATGCTGTTAAGCGCATGCGCCAGGAAGAAAGCCAGAAATGGGGATTCGACTTTGAACTGGAGACTCCCCTCCCAAGCTCTGCTGGATTCGTTTATGAAGTTATTCCAGAGAATTGTGTTCCGGAGTTCTACAGGTAATTGAATTTTATAAATTTTTCATAGTTATTTTACTAAACAGTTTCATTTTTCAGAACCAAAGTTCTCACTGTCAGAACCACATGCTCATCGCTGGACATCAGCTCAACGACTTTGACTCCATTGAGCTCTCCGAGCACATCTGATAAGGAGGAGCCCTCGCTGATGGATCCCAACAGCTCGTTCGAAGATGAAGAGGAACCGAAGAAGTGGCAATTCAGAGAGCCACCAACTCCACGGAAGACCCCAACAAAGCGTCAGCAGAAGATGACCGACTTCATGGCAGTTTCCCGTAAGAAGAATTCGTTGTCTCCAAACAAGCTGTCTCCGGTGAATGTGATCTTCACTCCAAAATCTCGTCGTCCAACGATCAGAACTCGATCTTCATGCTCTCCATACTAGAGGTTTCATTTTGACTTTTTTTTGCCCAATTCCACGGGGTTGAATCTAATCATTTGATTATCTCCTCGACAGTTTCTGAGTCTCTCTTAATTGTTCAACTAGTCATGTTTCCACAAATGTTTTATTGTTTGTTCCAAAAGCCCTGTGATCCATGTTTAGGAACTCTGTAACTCTTTTTTCCCATTGCCATTTGTTTTAAACAACTCAAAGAAAAATAAACCCTTTGAAATTATTTTAAGAACTGTATTCTGGTGTTTTCTTCAACTTATAAAAAAAAAGACGAATAGAAACTGGCACACGGTGCAGTTCCATTGGTAACTTCAGCAAAGAATATACTGAAATCACGAAAAGTGGTACAATTCCGCGCATAATTTTGAAACTTCTAACATTCTTCATTAACTTCAAACTTCAAACATTCTGTAAATGTTGTAAGATCAAATAAATCTTTCCCGGTTCACCCACTGCCACCCAAATAGACATTGCGCGATAACATGGTCATAGCTGTTTCCTGTGTG |
| Sequenced-based reagent | *oMS218 (5' HA block)* | This study | IDT | ttgtaaaacgacggccagtcgccggcattatgacaattttctgcgagagtttaaaatattacagatttttttaaattttgaaaaatatctaatattctcgaaaaattcgccttggaaaatttcgaaaaattcattttaaaaataggaaattcaaaattactactttagcattaaaaaaaatcataaaaaattctccaaatttttttagaagtttccaaaaaaaaaatcgcaaaaattaaatttgtggttttccaacaataaatggaccaaaatcaaaaatttccaccaaaaaaaaacataacttctcctcgaggagtacacgagctccgtaaatcgacacagacatttgtgaaaaaaattacttgaaaatcgtaaaatttcaacaaaaaaaaattctaattttttttccagATACTTCCGATTCACCGACAACAACATTGAAGCAATCAACGAGTTGCTCGATGAAGAGCTCCAAATTACTCAGAAAAAGATTGATGAGCAACGAAACACCCAAATTGCACAAATGAGCCAtCACCACACACCACGGCCATCCAAAGCAGCAAGATCTCTCAAATTTCATGGAGCATCGGGAGCCTCAGGAGCATCGATG |

*Continued on next page*

*Continued*

| Reagent type (species) or resource | Designation | Source or reference | Identifiers | Additional information |
|---|---|---|---|---|
| Sequenced-based reagent | *DHB-mNG-p2a* | This study | Twist Biosciences | CTTCCATTTCAGGTGTCGTGAACACGCTACCGGTCTCGAGAATTCACCGGATCCATGACAAATGATGTCACCTGGAGCGAGGCCTCTTCGCCTGATGAGAGGACACTCACCTTTGCTGAAAGATGGCAATTATCTTCACCTGATGGAGTAGATACAGATGATGATTTACCAAAATCGCGAGCATCCAAAAGAACCTGTGGTGTGAATGATGATGAAAGTCCAAGCAAAATTTTTATGGTGGGAGAATCTCCACAAGTGTCTTCCAGACTTCAGAATTTGAGACTGAATAATTTAATTCCCAGGCAACTTTTCAAGCCCACCGATAATCAAGAAACTGGTTCCGGGGCCCAGGGCAGCGGCATGGTGAGCAAGGGCGAGGAGGATAACATGGCCTCTCTCCCAGCGACACATGAGTTACACATCTTTGGCTCCATCAACGGTGTGGACTTTGACATGGTGGGTCAGGGCACCGGCAATCCAAATGATGGTTATGAGGAGTTAAACCTGAAGTCCACCAAGGGTGACCTCCAGTTCTCCCCCTGGATTCTGGTCCCTCATATCGGGTATGGCTTCCATCAGTACCTGCCCTACCCTGACGGGATGTCGCCTTTCCAGGCCGCCATGGTAGATGGCTCCGGATACCAAGTCCATCGCACAATGCAGTTTGAAGATGGTGCCTCCCTTACTGTTAACTACCGCTACACCTACGAGGGAAGCCACATCAAAGGAGAGGCCCAGGTGAAGGGGACTGGTTTCCCTGCTGACGGTCCTGTGATGACCAACTCGCTGACCGCTGCGGACTGGTGCAGGTCGAAGAAGACTTACCCCAACGACAAAACCATCATCAGTACCTTTAAGTGGAGTTACACCACTGGAAATGGCAAGCGCTACCGGAGCACTGCGCGGACCACCTACACCTTTGCCAAGCCAATGGCGGCTAACTATCTGAAGAACCAGCCGATGTACGTGTTCCGTAAGACGGAGCTCAAGCACTCCAAGACCGAGCTCAACTTCAAGGAGTGGCAAAAGGCCTTTACCGATGTGATGGGCATGGACGAGCTGTACAAGGGCAGCGGCGCCACCAACTTCAGCCTGCTGAAGCAGGCCGGCGACGTGGAGGAGAACCCCGGCCCCATGCCAGAGCCAGCGAAGTCTGCTCCCGCC |
| Sequenced-based reagent | *mScarlet-CAAX* | This study | Twist Biosciences | TCTAGAGGCAGCGGCCAGTGCACCAACTACGCCCTGCTGAAGCTGGCCGGCGACGTGGAGAGCAACCCCGGCCCCATGGTGAGCAAGGGCGAGGCAGTGATCAAGGAGTTCATGCGGTTCAAGGTGCACATGGAGGGCTCCATGAACGGCCACGAGTTCGAGATCGAGGGCGAGGGCGAGGGCCGCCCCTACGAGGGCACCCAGACCGCCAAGCTGAAGGTGACCAAGGGTGGCCCCCTGCCCTTCTCCTGGGACATCCTGTCCCCTCAGTTCATGTACGGCTCCAGGGCCTTCACCAAGCACCCCGCCGACATCCCCGACTACTATAAGCAGTCCTTCCCCGAGGGCTTCAAGTGGGAGCGCGTGATGAACTTCGAGGACGGCGGCGCCGTGACCGTGACCCAGGACACCTCCCTGGAGGACGGCACCCTGATCTACAAGGTGAAGCTCCGCGGCACCAACTTCCCTCCTGACGGCCCCGTAATGCAGAAGAAGACAATGGGCTGGGAAGCGTCCACCGAGCGGTTGTACCCCGAGGACGGCGTGCTGAAGGGCGACATTAAGATGGCCCTGCGCCTGAAGGACGGCGGCCGCTACCTGGCGGACTTCAAGACCACCTACAAGGCCAAGAAGCCCGTGCAGATGCCCGGCGCCTACAACGTCGACCGCAAGTTGGACATCACCTCCCACAACGAGGACTACACCGTGGTGGAACAGTACGAACGCTCCGAGGGCCGCCACTCCACCGGCGGCATGGACGAGCTGTACAAGTCCGGACTCAGATCTAAGCTGAACCCTCCTGATGAGAGTGGCCCCGGCTGCATGAGCTGCAAGTGTGTGCTCTCCTGACTAGAGTTAACATCGAGGGATCAAGCTTATCGATAATCAACCTCTGGATTACAAAATTTGT |

*Continued on next page*

*Continued*

| Reagent type (species) or resource | Designation | Source or reference | Identifiers | Additional information |
|---|---|---|---|---|
| Sequenced-based reagent | *H2B-mTurqoise2* | This study | Twist Biosciences | ATGCCAGAGCCAGCGAAGTCTGCTCCCGCCCCGAAAAAGGGCTCCAAGAAGGCGGTGACTAAGGCGCAGAAGAAAGGCGGCAAGAAGCGCAAGCGCAGCCGCAAGGAGAGCTATTCCATCTATGTGTACAAGGTTCTGAAGCAGGTCCACCCTGACACCGGCATTTCGTCCAAGGCCATGGGCATCATGAATTCGTTTGTGAACGACATTTTCGAGCGCATCGCAGGTGAGGCTTCCCGCCTGGCGCATTACAACAAGCGCTCGACCATCACCTCCAGGGAGATCCAGACGGCCGTGCGCCTGCTGCTGCCTGGGGAGTTGGCCAAGCACGCCGTGTCCGAGGGTACTAAGGCCATCACCAAGTACACCAGCGCTAAGGtTCCACCGGTCGCCACCATGGTGAGCAAGGGCGAGGAGCTGTTCACCGGGGTGGTGCCCATCCTGGTCGAGCTGGACGGCGACGTAAACGGCCACAAGTTCAGCGTGTCCGGCGAGGGCGAGGGCGATGCCACCTACGGCAAGCTGACCCTGAAGTTCATCTGCACCACCGGCAAGCTGCCCGTGCCCTGGCCCACCCTCGTGACCACCCTGTCCTGGGGCGTGCAGTGCTTCGCCCGCTACCCCGACCACATGAAGCAGCACGACTTCTTCAAGTCCGCCATGCCCGAAGGCTACGTCCAGGAGCGCACCATCTTCTTCAAGGACGACGGCAACTACAAGACCCGCGCCGAGGTGAAGTTCGAGGGCGACACCCTGGTGAACCGCATCGAGCTGAAGGGCATCGACTTCAAGGAGGACGGCAACATCCTGGGGCACAAGCTGGAGTACAACTACTTTAGCGACAACGTCTATATCACCGCCGACAAGCAGAAGAACGGCATCAAGGCCAACTTCAAGATCCGCCACAACATCGAGGACGGCGGCGTGCAGCTCGCCGACCACTACCAGCAGAACACCCCCATCGGCGACGGCCCCGTGCTGCTGCCCGACAACCACTACCTGAG<br>CACCCAGTCCAAGCTGAGCAAAGACCCCAACGAGAAGCGCGATCACATGGTCCTGCTGGAGTTCGTGACCGCCGCCGGGATCACTCTCGGCATGGACGAGCTGTACAAGTCTAGAGGCAGCGGCCAGTGCACCAACTACGCC |
| Sequenced-based reagent | *AID-link-miRFP670* | This study | Twist Biosciences | AATACAAGCTACTTGTTCTTTTTGCAGGATCCATCATCCCTTAATTAAGGATAGTGATTATCGATACATGAAGGAGAAGAGTGCTTGTCCTAAAGATCCAGCCAAACCTCCGGCCAAGGCACAAGTTGTGGGATGGCCACCGGTGAGATCATACCGGAAGAACGTGATGGTTTCCTGCCAAAAATCAAGCGGTGGCCCGGAGGCGGCGGCGTTCGTGAAGGTATCAATGGACGGAGCACCGTACTTGAGGAAAATCGATTTGAGGATGTATAAAGGTGCTAGCGGTGCAGGCGCCATGGTAGCAGGTCATGCCTCTGGCAGCCCCGCATTCGGGACCGCCTCTCATTCGAATTGCGAACATGAAGAGATCCACCTCGCCGGCTCGATCCAGCCGCATGGCGCGCTTCTGGTCGTCAGCGAACATGATCATCGCGTCATCCAGGCCAGCGCCAACGCCGCGGAATTTCTGAATCTCGGGAAGCGTACTCGGCGTTCCGCTCGCCGAGATCGACGGCGATCTGTTGATCAAGATCCTGCCGCATCTCGATCCCACCGCCGAAGGCATGCCGGTCGCGGTGCGCTGCCGGATCGGCAATCCCTCTACGGAGTACTGCGGTCTGATGCATCGGCCTCCGGAAGGCGGGCTGATCATCGAACTCGAACGTGCCGGCCCGTCGATCGATCTGTCAGGCACGCTGGCGCCGGCGCTGGAGCGGATCCGCACGGCGGGTTCACTGCGCGCGCTGTGCGATGACACCGTGCTGCTGTTTCAGCAGTGCACCGGCTACGACCGGGTGATGGTGTATCGTTTCGATGAGCAAGGCCACGGCCTGGTATTCTCCGAGTGCCATGTGCCTGGGCTCGAATCCTATTTCGGCAACCGCTATCCGTCGTCGACTGTCCCGCAGATGGCGCGGCAGCTGTACGTGCGGCAGCGCGTCCGCGTGCTGGTCGACGTCACCTATCAGCCGGTGCCGCTGGAGCCGCGGCTGTCGCCGCTGACCGGGCGCGATCTCGACATGTCGGGCTGCTTCCTGCGCTCGATGTCGCCGTGCCATCTGCAGTTCCTGAAGGACATGGGCGTGCGCGCCACCCTGGCGGTGTCGCTGGTGGTCGGCGGCAAGCTGTGGGGCCTGGTTGTCTGTCACCATTATCTGCCGCGCTTCATCCGTTTCGAGCTGCGGGCGATCTGCAAACGGCTCGCCGAAAGGATCGCGACGCGGATCACCGCGCTTGAGAGCTAA |
| Chemical compound, drug | Carbenicillin | Alfa Aesar | #J61949 | |
| Chemical compound, drug | Isopropyl-$\beta$-D-thiogalactoside | Thermo Scientific | #R0393 | |
| Chemical compound, drug | Palbociclib | MedChemExpress | #HY-A0065 | |

*Continued on next page*

*Continued*

| Reagent type (species) or resource | Designation | Source or reference | Identifiers | Additional information |
|---|---|---|---|---|
| Chemical compound, drug | Hygromycin B | Millipore | #400052 | |
| Chemical compound, drug | Sodium Azide | Sigma-Aldrich | #S2002 | |
| Chemical compound, drug | Tricaine | Sigma-Aldrich | #E10521 | *C. elegans* |
| Chemical compound, drug | Tricaine | Pentair | #TRS1 | *D. rerio* |
| Chemical compound, drug | Levamisole | Sigma-Aldrich | #L9756 | |

[*]The ZF degron in CDT-1::ZF::GFP does not cause degradation, because the *zif-1(gk117)* null allele removes the E3 ligase component ZIF-1 that recognizes the ZF tag (*Sallee et al., 2018*).

## *C. elegans* transgenic strain generation

Transgene insertion was performed via CRISPR/Cas9 genome engineering to generate single copy knock-ins to a known neutral locus on chromosome I or II using a self-excising cassette (SEC)-based method (*de la Cova et al., 2017*; *Dickinson et al., 2015*). Homologous repair templates and guide plasmids were graciously provided by Bob Goldstein, targeting the MosSCI integration sites ttTi4348 and ttTi5605 on chromosome I and II, respectively. CRISPR microinjection products were prepared using the PureLink HQ Mini Plasmid DNA Purification Kit from Invitrogen (K210001). An additional wash step was included prior to the final ethanol wash, using 650 µL of 60% 4 M guanidine hydrochloride (Fisher Scientific, BP178-500; pH 6.5, 40% isopropanol) yielding a marked increase in knock-in efficiency. All purified microinjection products were stored at 4 °C.

Injection mixes were freshly made before each round of injection. These mixes contain Cas9-sgRNA plasmids (50 ng/µL), homologous repair templates (50 ng/µL), and a co-injection marker (pCFJ90, 2.5 ng/µL). Injection mixes were injected into the gonads of young adult *C. elegans* N2 hermaphrodites. Successful integrants were identified in the F3 offspring of injected worms (*Dickinson et al., 2015*). Injected young adult hermaphrodites of the relevant parent strain were then each individually transferred to a fresh OP50 plate and allowed to lay eggs for three days at 25 °C. On day 3, 400 µL of a 5 mg/mL stock of hygromycin B (Millipore, 400052) was added to the plates to a final plate concentration of 0.25 mg/mL. After five days of hygromycin B exposure, surviving dominant *sqt-1* roller (Rol) worms were singled out onto fresh OP50 plates, checked for expression of the desired transgene/genomic edit and the presence of extrachromosomal array markers on a fluorescence dissecting microscope (frame and automation: Zeiss Axio Zoom.V16, light source: Lumencor SOLA light engine). The Rol phenotype was assessed for Mendelian inheritance, and if possible, the genomic edit was homozygosed. Once homozygosed, selectable markers (hygromycin B resistance and dominant *sqt-1* Rol phenotype) were removed from the genome using heat shock-inducible Cre-Lox recombination via either a 3–4 hr heat shock at 34 °C or overnight (8–12 hr) heat shock of large numbers of L1 and L2 stage animals at 26 °C in an air incubator. After two days, wild type worms were singled out one to a plate and progeny assessed for expression and homozygosity of the desired genomic insertion.

## Zebrafish transgenic line generation

Three transgenic lines were generated, including *Tg(ubb:Lck.mNeonGreen)*[sbu107], *Tg(hsp70l:DHB.mNeonGreen-p2a-H2B.mScarlet)*[sbu108], and *Tg(hsp70l:DHB.mScarlet-p2a-H2B.miRFP670)*[sbu109]. These lines were created using the Tol2 transposable element system (*Kawakami, 2004*). Zebrafish plasmids for generating transgenic lines were created using a *tol2* plasmid vector containing the *hsp70l* promoter based on previous plasmids constructs (*Row et al., 2016*). For the *hsp70l:DHB.mNeonGreen-p2a-H2B.mScarlet* plasmid, Gibson cloning was used to insert DNA encoding amino acids 994–1087 of human DHB fused to the N-terminus of mNeonGreen, followed by the P2A viral peptide sequence and human H2B with a C-terminal mScarlet fusion. The same method was used to

generate *hsp70l:DHB.mScarlet-p2a-H2B.miRFP670*, except mScarlet and miRFP670 were used instead of mNeonGreen and mScarlet, respectively. The *tol2 hsp70l* vector was also used to create the *ubb:Lck.mNeonGreen* plasmid. The *hsp70l* promoter was replaced with the *ubb* promoter (*Mosimann et al., 2011*), followed by mNeonGreen with an N-terminal membrane targeting sequence from *Mus musculus* LCK (amino acids MGCVCSSNPE). Each plasmid was co-injected with in vitro transcribed *tol2* transposase mRNA. One nanoliter of injection mix containing 25 pg/nl of plasmid and 25 pg/nl of tol2 mRNA were injected into wild type zebrafish embryos at the 1 cell stage. Injected embryos were raised to adults and screened for germline transmission.

## Zebrafish mosaic analysis

One nanoliter of injection mix containing 25 pg/nl of HS-PCNA-GFP and 25 pg/nl of tol2 mRNA were injected into *Tg(hsp70l:DHB.mScarlet-p2a-H2B.miRFP670)*[sbu109] zebrafish embryos at the 1 cell stage.

## Molecular biology

Synthetic DNAs were ordered as gBlocks from Integrated DNA Technologies (IDT) or gene fragments from Twist BioScience (see Key Resources Table). The nucleotide sequence of DHB (index 1.0) was codon-optimized for *C. elegans* somatic expression and the P2A sequence used in pWZ193 (index 0.2; see KRT) de-optimized to increase the efficiency of ribosome stalling (*Lo et al., 2019*; *Redemann et al., 2011*). The *C. elegans rps-0* and *rps-27* promoters and the *pcn-1* promoter and coding sequence were all amplified from N2 genomic DNA. Sequences of all primers and synthetic DNAs are provided in the KRT. Synthetic gene fragments and amplified DNAs were cloned via Gibson Assembly (*Barnes, 1994*; *Gibson et al., 2010*; *Gibson et al., 2009*) or NEBuilder HiFi into target plasmids.

Constructs used for zebrafish transgenes were made from PCR products amplified from synthetic Twist BioScience gene fragment sequences followed by NEBuilder HiFi cloning. Human DHB and H2B sequences were used for making the DHB transgenes, and the human membrane targeting Lck sequence was used for the *ubb:Lck.mNeonGreen* transgene. All primers and synthetic gene fragment sequences are available in the KRT.

## Microinjection setup

Microinjections for *C. elegans* transgenesis were performed on an injection setup combining a Zeiss Axio Observer A1 inverted compound frame, EC Plan-Neofluar 40x/0.75 NA DIC objective and floating stage, with a Narashige manual micromanipulator and a picoliter injection system from Warner for fine control of delivered volume. Microinjection needles (Sutter) were pulled on a Sutter P-97 reconditioned and calibrated by Sutter.

Zebrafish microinjections were performed on either a Leica S6e or a Zeiss Stemi 508 dissecting microscope using a Narishige manual micromanipulator and a Warner picoliter injecting system. Glass needles were pulled on a P-1000 puller from Sutter Instruments.

## *C. elegans* RNAi perturbations

RNAi was delivered by feeding *E. coli* strain HT115(DE3) expressing double-stranded RNA (dsRNA) to synchronized L1 stage strains. Transcription of dsRNA was induced with 1 mM isopropyl-b-D-1-thiogalactopyranoside (Thermo Scientific, R0393) in bacterial cultures for 1 hr at 37 °C. After an hour, cultures were plated on NGM plates topically treated with 2.5 µl each of 30 mg/mL carbenicillin (Alfa Aesar, J61949) and 10 µl of 1 M IPTG. The RNAi vector targeting *cdk-1* was obtained from the Vidal RNAi library (*Rual et al., 2004*). The empty vector L4440 was used as a negative control. RNAi vectors were verified by Sanger sequencing.

## *C. elegans* CKI-1 experiments

For heat shock CKI-1 experiments, the following strains were used DQM406 (*hsp*>CKI-1::BFP; *rps-0*>DHB::mKate2) and DQM394 (*rps-0*>DHB::mKate2). Synchronized L1 animals were plated on OP50 and allowed to develop to mid-L3. Plates were then placed at 30 °C in an air incubator for 3 hr. Animals were then placed at 20 °C and allowed to recover from heat shock for 20–40 min before being mounted for static imaging.

For assessing endogenous CKI-1 levels in *Figure 4*, strain DQM586 (GFP::CKI-1; *rps-27*>DHB::2xmKate2) was utilized. Briefly, L1 animals were synchronized via sodium hypochlorite treatment and plated on OP50 at 25 °C and analyzed at the P6.p 2 cell, 4 cell, and 8 cell stages. DQM586 was superficially wild type, but several phenotypes, revealed by confocal microscopy and/ or analyzed in this study (e.g. the presence of larger somatic cells than normal in the L3 and L4 stages, including the anchor cell), led to the conclusion that the N-terminal GFP fusion (which lacks a flexible linker) resulted in animals displaying a gain-of-function effect of GFP::CKI-1. Early cell cycle exit in the VPCs was determined by lineage analysis of each image and comparing the size of individual VPCs in GFP::CKI-1 animals to wild type. A VPC was considered to have undergone early cell cycle exit if it failed to divide (larger nucleus than normal) and showed strong nuclear localization of DHB::2xmKate2 consistent with a CDK$^{low}$ state.

## Zebrafish drug perturbations

Palbociclib (PD-0332991), a selective inhibitor of CDK4/6, was purchased from MedChemExpress (HY-A0065). A 5 mM stock solution in embryo media was prepared and stored at −80 °C for up to six months. Prior to each experiment, palbociclib was thawed and diluted in embryo media to a final concentration of 50 μM. Control experiments were performed by treating zebrafish embryos with embryo media only. Embryos were placed in palbociclib at 16 somites for 5 hr at 22 °C. DHB measurements were performed blinded to avoid bias.

## Microscopes for Live-Cell imaging

All live-cell imaging of *C. elegans* and zebrafish, unless indicated otherwise, was performed on a custom-assembled spinning disk confocal microscope consisting of a Zeiss Axio Imager A2 frame, a Borealis modified Yokogawa CSU10 spinning disc, an ASI 150-micron piezo stage controlled by a MS2000, an ASI filter wheel and a Hamamatsu ImagEM X2 EM-CCD camera. The imaging objective used for *C. elegans* imaging was a Plan Apochromat 100x/1.4 NA DIC objective (Carl Zeiss). For zebrafish imaging, a Plan Apochromat 63x/1.0 NA water dipping objective (Carl Zeiss) was used. L3 stage *C. elegans* larvae shown in *Figure 1D and D'* were imaged on a separate custom-assembled spinning disk confocal microscope consisting of an automated Zeiss frame, a Yokogawa CSU10 spinning disc, a Ludl stage controlled by a Ludl MAC6000 and an ASI filter turret attached to a Photometrics Prime 95B camera. The imaging objective used was a Plan Apochromat 63x/1.4 NA DIC objective (Carl Zeiss). For both aforementioned microscopes, laser illumination was provided by a six-line, 405/442/488/514/561/640 nm Vortran laser merge driven by a custom Measurement Computing Microcontroller integrated by Nobska Imaging, Inc Both microscopes were controlled with Metamorph software (version: 7.10.2.240) and laser power levels were set with Vortran's Stradus VersaLase eight software. In *Figure 1—figure supplement 1A*, live imaging of *C. elegans* embryos was performed on a Nikon Ti-E inverted microscope using a Plan Apochromat 60x/1.4 NA oil immersion objective and controlled by Nikon's NIS-Elements software (version: 4.30). Images were acquired with an Andor Ixon Ultra back thinned EM-CCD camera using 488 nm or 561 nm imaging lasers and a Yokogawa X1 confocal spinning disk head equipped with a 1.5 Å magnifying lens. For time-lapse imaging of the *C. elegans* germline and embryos in *Figure 1C* and *Figure 1—figure supplement 1C* and *Figure 1—video 1*, recordings were acquired using a Yokogawa CSUW1 SoRa spinning disk confocal in SoRa disk mode with 1.0x relay lens, a 60x/1.27 NA water immersion objective and a Prime 95B sCMOS camera mounted on a Nikon Ti-2 stand. Nikon's NIS-Elements software (version: 4.3) was used for image acquisition.

## *C. elegans* imaging conditions

For static imaging experiments, worms were anesthetized by mounting on a 7.5% noble agar pad containing sodium azide (Sigma-Aldrich, S2002) (*Martinez and Matus, 2020*; *Matus et al., 2015*). Time-lapse imaging of *C. elegans* was performed using a modified version of a previously published protocol (*Kelley et al., 2017*). We substituted in a 24 mm square coverslip #1.5 (Fisher Scientific, 12–541-B) and divided the imaging agar pad into two asymmetric smaller portions (each 2–3 mm squares), filling the void space under the coverslip with 5 mM levamisole in M9 buffer or M9 buffer alone. These modifications allowed for much longer imaging durations and substantially reduced

sample Z-drift over the course of the imaging session on both upright and inverted microscope systems.

Anesthesia was performed in a spot dish in ~50 µl of a 0.1% tricaine (Sigma-Aldrich, E10521)/ 0.01% levamisole hydrochloride (Sigma-Aldrich, L9756) anesthetic (*Kirby et al., 1990*; *Maddox and Maddox, 2012*; *Wong et al., 2011*). For some experiments, this tricaine-levamisole solution was substituted for 5 mM levamisole in M9 buffer. When levamisole was used alone, to maintain animals in an anesthetized state for long-duration time-lapse imaging, imaging chambers were flooded with 5 mM levamisole in M9 instead of M9.

Embryos for imaging (*Figure 1C*, *Figure 1—figure supplement 1A*) were collected by dissection from gravid hermaphrodites and incubated for 4–4.5 hr in M9 at room temperature (*Figure 1—figure supplement 1A*) or imaged immediately (*Figure 1C*, *Figure 1—video 1*). For live imaging, images were taken at a sampling rate of 0.5 µm. For time-lapse, z-stacks were collected every four (*Figure 1—figure supplement 1A*) or three min (*Figure 1C*, *Figure 1—video 1*). For time-lapse of the germline (*Figure 1—figure supplement 1C*, *Figure 1—video 1*), young adult worms were lightly immobilized using 0.1 mM levamisole in M9 buffer and mounted on 5% agarose pads.

## Zebrafish imaging conditions

Zebrafish were mounted in a 35 mm glass bottom dish with uncoated #1.5 coverslip and 20 mm glass diameter (MatTek). A thin layer of 1% agarose dissolved in embryo media (*Westerfield, 2007*), was added to the dish covering the glass bottom. Once solidified, a P10 pipette tip was used to punch holes in the agarose. Embryos were added to 1% low melting point agarose dissolved in embryo media containing 1x tricaine (24x stock 0.4 g/l; Pentair, TRS1), and then one embryo was added to each of the punched holes. Embryos were manipulated gently with an eyelash while the agarose solidified to ensure proper orientation. For 72 hpf embryos, animals were anesthetized in 1x tricaine prior to mounting in 1% low melt agarose with 1x tricaine. In all cases imaging dishes were filled with embryo media containing 1x tricaine.

## Image processing

Hand quantification of images was performed in Fiji (version: 2.0.0-rc-69/1.52 p) (*Schindelin et al., 2012*). Due to the high level of amplifier noise in EM-CCD images, and to remove any remaining out-of-focus fluorescence in these confocal micrographs, a rolling ball background subtraction was used (size = 50) (*Sternberg, 1983*). After a recording was qualified for inclusion, ratiometric measurements were obtained.

First, the Z plane containing the center of the cell of interest was located. Using the freehand tool, a conservative toroid was drawn around the nucleus and excluding the nucleolus if present, which does not localize the CDK sensor. The fluorescent histone and corresponding DIC and DHB images were used to assess the accuracy of this toroid. A measurement of mean gray value was obtained. Then, a region of perinuclear cytoplasm was chosen, avoiding pixels belonging to the cytoplasm of neighboring cells. The mean gray value of the cytoplasmic patch was then measured. These values were recorded and a cytoplasmic: nuclear ratio was calculated. If there were multiple cells of interest in the image, the procedure was repeated for each cell. For time-lapse recordings, this procedure was repeated at each time point.

## Statistical analyses

To evaluate the predictability of CDK activity (readout as the ratio of cytoplasmic-to-nuclear intensity of DHB) on proliferative versus quiescent cell fate in different cell-cycle phases, we created a receiver operating characteristic (ROC) curve for CDK activity at each time point relative to anaphase. Using the *perfcurve* function in MATLAB, we calculated the area under the curves (AUC) as the indicator of predictability. We then built a classifier to predict proliferative vs. quiescent cell fates based on CDK activity after anaphase. For each time point, we chose the CDK-activity threshold for classification that maximizes the geometric mean of specificity (1 – false positive rate) and sensitivity (true positive rate). We tested the classifier in a second dataset, the stochastic division of the vulval D cell (see *Figure 5*). To predict the cell fate of each trace, we made independent classifications on each relevant time point based on CDK activity and use the majority class of all relevant time points as the classification for the trace. For traces recorded beyond 60 min after anaphase, we used all time points after

60 min post-anaphase, since these time points allow near-perfect prediction (AUC>0.9). For traces recorded beyond 20 min but within 60 min after anaphase, we used their last three time points, since these time points show good and increasing prediction power with AUC>0.8.

Bootstrapping was performed in MATLAB R2019A. The code used is available at GitHub (https://github.com/abraham-kohrman/matus-dhb-stats; *Kohrman, 2020*; copy archived at swh:1:rev:9c88bc74fa1ca0793b2ee9598d1842a482581400). Custom code for statistical testing may not be compatible with MATLAB releases older than R2019A and may require the use of MATLAB Toolboxes. Briefly, when single timepoint samples did not exhibit normal distributions, empirical statistics were calculated. For single timepoint experiments, a bootstrapped distribution of the difference between mean groups was calculated for each comparison (*Equation 1*).

$$|\hat{x}_1 - \hat{x}_2| \tag{1}$$

$10^8$ statistical simulations were performed by random sampling without replacement in MATLAB. A *p*-value was calculated by determining the proportion of simulated differences with values greater than the true difference.

For comparisons of time course data, a mean trend line was calculated for each dataset to be compared. The area between the mean trend lines was calculated. In MATLAB, this was performed as the sum of the absolute value of the difference at each time point. Where $x_1$ corresponds to the first trend line and $x_2$ corresponds to the second trend line.

$$\|x_1 - x_2\|_1 \tag{2}$$

Statistical simulations were performed by random partitioning of the data without replacement into two groups with the same sizes as the original groups. Mean trend lines were then calculated for these randomly assigned groups, and as before the statistic was calculated. $10^8$ simulated replicates were performed to estimate the distribution of the difference statistic. In a manner analogous to bootstrapping, *p*-value was calculated by determining the proportion of simulations with more extreme statistical values than the observed statistic. See *Figure 8—figure supplement 1* for a detailed schematic of the procedure.

## Reporting of statistical results

The $\alpha$ value for this study was nominally 0.05, however exact *p*-values and *n* (number of cells) are reported in all cases. When no simulation produced a more extreme result than the true data configuration, *p*-values are reported as $p<1\times10^{-7}$, rather as the true probability value is so small, as to be outside the range of accurately calculable probability values. For every comparison performed, plots of distributions of empirically calculated statistics are available upon request.

To interpret *p*-values as presented, it is important to note our null hypothesis which can be formulated as: The categorization (e.g. into C lineage vs. D lineage cells or treated vs untreated cells) is not better than random. In short, the *p*-values we have corresponded to the probability that the difference between the mean or mean trend lines arose by chance. Another formulation would be the odds that the categorization of the data is meaningless. Throughout the study, an $\alpha$ value of 0.05 is used for significance. A *p*-value of 0.05 corresponds to the statement that 95% of random reassortments of the data yielded a difference between the means/mean trend lines less extreme than the true, observed difference.

In the course of data collection for this manuscript, many animals were recorded that were not included in this manuscript. In order to be considered for analysis, recordings had to satisfy the following criteria: (1) a cell of interest had to have been present in the recording, (2) the cell of interest must have exhibited at least one anaphase during the recording, and (3) the animal must have appeared phenotypically normal at the beginning and end of the recording. Additional criteria for exclusion were the presence of a stalled metaphase plate at any point in the Video or unexpected developmental arrest.

## Computational resources

For data analysis, two workstation computers were used. Both systems boot into Windows 10 (Microsoft) off a 1 TB M.2 drive (Samsung 970 EVO Plus). The first system consists of an I9-9900X processor (Intel), a GeForce GTX 1070 Ti GPU (Nvidia) and 128 GB of DDR4 RAM (Corsair). The

second system has an I9-9900K processor (Intel), a GeForce RTX 2070 GPU (Nvidia) and 64 GB of DDR4 RAM (G.Skill Ripjaws). Data were stored on a 4 TB RAID0 array consisting of two 2 TB drives (Samsung) and a 2 TB RAID0 array consisting of two 1 TB Drives (Samsung), respectively. System integration, support and maintenance performed by Nobska Imaging, Inc.

### Generation of figures and videos

Data for figures were plotted in GraphPad Prism (version: 8.1.2). Micrographs in all figures were reviewed and selected in Fiji. Figure micrographs were contrast and brightness adjusted for ease of display in Adobe Photoshop CC (version: 20.0.6) or Fiji. Figures were assembled in Adobe Illustrator CC (version: 23.0.26). Supplemental Videos were selected in Fiji and clipped to the desired length. The plane of interest was selected, and a time-lapse montage of channels was created. Time-lapse Videos were rotated to standard orientation, cropped to the relevant region and timestamps and scale bars annotations were added. Brightness and contrast were adjusted for ease of viewing. Videos showing more than one channel were assembled using the multi-stack montage plugin (https://github.com/BIOP/ijp-multi-stack-montage).

## Acknowledgements

We thank O. Quintero and the Fall 2020 BIO 317 students along with D Özpolat, J Ward, D Pisconti, C-K Hu and MJ Gacha-Garay for helpful feedback and comments on the manuscript; MJ Gacha-Garay, N Bhattacharji and S Flanagan for fish care; J Maghakian and L Yang for consultation on statistical modeling; D Dickinson and B Goldstein for assistance with the CRISPR single copy knock-in strategy; and T Geer of Nobska Imaging, Inc for helping maintain our spinning disk confocal microscopes. This work was funded by the NIH NIGMS [1R01GM121597-01 to DQM and 1R01GM124282 to BLM]. DQM and BLM are both Damon Runyon-Rachleff Innovators supported by the Damon Runyon Cancer Research Foundation [DRR-47–17]. BLM also received support from the NSF [IOS 1452928] and the Pershing Square Sohn Cancer Research Alliance. RCA, AQK, JJS and MAQM are all supported by the NIGMS [1F32133131-01, F31GM128319-01, 3R01GM121597-02S1/S2, respectively]. TNM-K is supported by the NIH NICHD [F31HD100091-01]. NJP is supported by the ACS [132969-PF-18-226-01-CSM]. MDS is supported by the NIH NIGMS [1K99GM13548901]. JLF and SLS are both supported by an NIH Director's New Innovator Award (DP2GM1191136-01 and DP2-CA238330, respectively). SLS is also supported by an ACS Research Scholar Grant (RSG-18-008-01), a Pew-Stewart Scholar Award, a Beckman Young Investigator Award, a Boettcher Webb-Waring Early-Career Investigator Award, a Kimmel Scholar Award (SKF16-126), and a Searle Scholar Award (SSP-2016–1533). Some strains were provided by the *Caenorhabditis* Genetics Center, which is funded by the NIH ORIP [P40 OD010440].

## Additional information

### Funding

| Funder | Grant reference number | Author |
|---|---|---|
| National Institutes of Health | 1R01GM121597 | David Q Matus |
| National Institutes of Health | 1R01GM124282 | Benjamin L Martin |
| Damon Runyon Cancer Research Foundation | DRR-47-17 | Benjamin L Martin David Q Matus |
| National Science Foundation | IOS 1452928 | Benjamin L Martin |
| Pershing Square Sohn Cancer Research Alliance | | Benjamin L Martin |
| National Institutes of Health | 1F32133131 | Rebecca C Adikes |
| National Institutes of Health | F31GM128319 | Abraham Q Kohrman |
| American Cancer Society | 132969-PF-18-226-01-CSM | Nicholas J Palmisano |
| National Institutes of Health | F31HD1000091 | Taylor N Medwig-Kinney |

| National Institutes of Health | DP2GM1191136 | Sabrina L Spencer |
| National Institutes of Health | DP2-CA238330 | Jessica L Feldman |
| American Cancer Society | RSG-18-008-01 | Sabrina L Spencer |
| Pew Charitable Trusts | | Sabrina L Spencer |
| Boettcher Foundation | | Sabrina L Spencer |
| Searle Scholars Program | SSP-2016-1533 | Sabrina L Spencer |
| National Institutes of Health | 1K99GM13548901 | Maria D Sallee |
| National Institutes of Health | 3R01GM121597-03S1 | Michael AQ Martinez |
| National Institutes of Health | 3R01GM121597-02S1 | Jayson J Smith |

The funders had no role in study design, data collection and interpretation, or the decision to submit the work for publication.

## Author contributions

Rebecca C Adikes, Conceptualization, Data curation, Formal analysis, Supervision, Investigation, Visualization, Methodology, Writing - original draft, Writing - review and editing; Abraham Q Kohrman, Conceptualization, Data curation, Software, Formal analysis, Investigation, Visualization, Methodology, Writing - original draft, Writing - review and editing; Michael A Q Martinez, Conceptualization, Formal analysis, Investigation, Visualization, Methodology, Writing - original draft, Writing - review and editing; Nicholas J Palmisano, Taylor N Medwig-Kinney, Formal analysis, Investigation, Visualization, Writing - review and editing; Jayson J Smith, Maria D Sallee, Formal analysis, Investigation, Writing - review and editing; Mingwei Min, Jessica L Feldman, Formal analysis, Writing - review and editing; Ononnah B Ahmed, Robert D Morabito, Nicholas Weeks, Investigation; Nuri Kim, Simeiyun Liu, Formal analysis, Investigation; Qinyun Zhao, Software, Formal analysis; Wan Zhang, Supervision, Investigation; Michalis Barkoulas, Investigation, Writing - review and editing; Ariel M Pani, Data curation, Formal analysis, Investigation, Writing - review and editing; Sabrina L Spencer, Supervision, Investigation, Writing - review and editing; Benjamin L Martin, Conceptualization, Data curation, Formal analysis, Supervision, Funding acquisition, Investigation, Visualization, Methodology, Writing - original draft, Project administration, Writing - review and editing; David Q Matus, Conceptualization, Data curation, Formal analysis, Supervision, Funding acquisition, Validation, Investigation, Visualization, Methodology, Writing - original draft, Project administration, Writing - review and editing

## Author ORCIDs

Rebecca C Adikes https://orcid.org/0000-0002-7526-8701
Abraham Q Kohrman https://orcid.org/0000-0002-3726-1090
Michael A Q Martinez https://orcid.org/0000-0003-1178-7139
Nicholas J Palmisano https://orcid.org/0000-0002-7992-4462
Jayson J Smith https://orcid.org/0000-0001-8525-7873
Taylor N Medwig-Kinney http://orcid.org/0000-0001-7989-3291
Mingwei Min http://orcid.org/0000-0002-9050-5330
Jessica L Feldman http://orcid.org/0000-0002-5210-5045
Michalis Barkoulas http://orcid.org/0000-0003-1974-7668
Sabrina L Spencer https://orcid.org/0000-0002-5798-3007
Benjamin L Martin https://orcid.org/0000-0001-5474-4492
David Q Matus https://orcid.org/0000-0002-1570-5025

## Ethics

Animal experimentation: This study was performed in strict accordance with the recommendations in the Guide for the Care and Use of Laboratory Animals of the National Institutes of Health. All of the animals were handled according to approved institutional animal care and use committee (IACUC) protocols (#2012-1932 - R2 - 1.15.21- FI) of Stony Brook University. The protocol was approved by the Office of Research Compliance of Stony Brook University.

### Decision letter and Author response
Decision letter https://doi.org/10.7554/eLife.63265.sa1
Author response https://doi.org/10.7554/eLife.63265.sa2

## Additional files

### Supplementary files
• Supplementary file 1. Summary of statistical analyses. Associated *p*-values reported for all statistical analyses performed. See *Figure 8—figure supplement 1* and Materials and methods for additional details of statistical analyses.

• Transparent reporting form

### Data availability
All data generated or analysed during this study are included in the manuscript and supporting files.

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
