## [Decision Letter]

**Acceptance summary:**

This paper describes the generation and analysis of live cell cycle reporters in two important experimental model systems, *C. elegans* and zebrafish. These tools represent significant advances over currently available live cell cycle reporters in that they distinguish between G0 and G1 phases, allowing direct visualization of the proliferation/ quiescence decision in vivo. We anticipate that these tools will be of great value to the wider scientific community.

**Decision letter after peer review:**

Thank you for submitting your article "Visualizing the metazoan proliferation-terminal differentiation decision in vivo" for consideration by *eLife*. Your article has been reviewed by three peer reviewers, one of whom is a member of our Board of Reviewing Editors, and the evaluation has been overseen by Didier Stainier as the Senior Editor. The following individual involved in review of your submission has agreed to reveal their identity: Paul W Sternberg (Reviewer #2).

The reviewers have discussed the reviews with one another and the Reviewing Editor has drafted this decision to help you prepare a revised submission.

We would like to draw your attention to changes in our revision policy that we have made in response to COVID-19 (https://elifesciences.org/articles/57162). Specifically, we are asking editors to accept without delay manuscripts, like yours, that they judge can stand as *eLife* papers without additional data, even if they feel that they would make the manuscript stronger. Thus the revisions requested below largely address clarity and presentation or can be addressed by re-analyzing existing data.

Summary:

In this manuscript, Adikes et al., describe the generation and analysis of CDK cell cycle reporters in *C. elegans* and zebrafish. The reporters are based on a well characterized reporter constructed a number of years ago by Spencer and colleagues and used to great effect to measure the proliferation/quiescence decision in mammalian cells grown in culture. While the previous cell culture analyses were interesting and conceptually useful, it was unknown whether these finding were applicable to cell cycle dynamics in vivo. Thus, the reviewers were excited to see the further development of these tools for use in two different, important experimental animal systems. The authors convincingly demonstrated the efficacy of these reporters within the essentially invariant cell lineages of *C. elegans*, and most strikingly, took advantage of cases where the lineages are not quite invariant.

These tools represent significant advances over currently available live cell cycle reporters (such as FUCCI) in that they distinguish between G0 and G1 phases, allowing direct visualization of the proliferation/ quiescence decision in vivo. Furthermore, dynamic sensor levels are highly predictive of cell cycle exit. Overall, this is a beautiful, convincing, and detailed study. The writing is thorough and cogent, the design of the reporters is elegant, the data are of high quality and well quantified, and – most importantly – the reporters seem to work. Reviewers anticipate that these tools will be of great value to the wider scientific community, and recommend publishing this study once the following concerns are addressed.

Essential revisions:

1) Throughout the manuscript, the authors conflate being predictive with being determinative, as demonstrated in the legend of Figure 8: "CDK activity shortly after mitotic exit determine future cell fate". Even if the correlation between CDK^inc^ / continued cycling and CDK^low^ / quiescence is 100%, it does not necessarily mean that CKI/CDK activity is instructive (i.e. determinative) for entry into quiescence. CKI activity may not even be required, as there could be other, redundant inputs into cell cycle exit. For this reason, we recommend re-wording some results and conclusions, taking care not to be too strident in concluding mechanism of quiescence and "cell fate" from the reporter analyses.

2) The authors often equate quiescence after cell cycle exit with terminal differentiation, as in the first sentence of the conclusion: "[the sensor] can distinguish between proliferative and terminally differentiated cells within an hour of cell birth". The sensor is highly predictive of exit from the cell cycle and thus likely a quiescent state, but not necessarily terminal differentiation, as cells in many different species can be quiescent without terminally differentiating. Again, the reviewers recommend simply taking care with the language and sticking close to what the sensor is actually measuring, which will in no way diminish the value of the tools or the study. We also recommend changing the title of this study to reflect this distinction, from "proliferation-terminal differentiation decision" to "proliferation-quiescence decision", or something similar.

3) On a related note, the finding that the vast majority of zebrafish tail bud cells at 24 hpf are "terminal" is surprising, and seemingly at odds with the many tail bud cells in Figure 7A with high cytoplasmic DHB levels and presumed to be in G2, and with supplementary Figure 7—figure supplement 1D, where 50% of cells are shown to be in G1.

4) In zebrafish embryos/larvae, the authors compare DHB ratios in progenitor cells at 24 hpf and with differentiated cells at a significantly later time-point: 72 hpf (Figure 7). When and how often are transgenic zebrafish embryos heat-shocked to induce DHB sensor expression? If embryos examined at 24 hpf and 72 hpf were heat-shocked at the same point in development, could reduced cytoplasmic levels of DHB at 72 hpf compared with 24 hpf be explained by gradual degradation of the DHB sensor? To ensure that differences in DHB ratios reflect the distinction between G1 and G0 rather than differences in developmental age, we recommend that these comparisons be made between differentiated and undifferentiated cells within embryos of the same stage.

5) In *C. elegans*, the authors showed that each tissue exhibits a distinct range of DHB ratios throughout the cell cycle. In other words, that DHB ratios cannot necessarily be directly compared between different tissues. However, comparisons in zebrafish between epidermal cells at 72 hpf and notochord progenitor cells (Figure 7H) are used to illustrate the differences in DHB sensor ratios between G1 and G0. Given the tissue-specific nature of the DHB ratios, these data would be more convincing if sensor levels were compared between cells of the same tissue type (for example, notochord progenitors with differentiated notochord.)

6) The predictive model appears to have been trained using the VPC lineage and is shown to predict behavior only of VPC cells. Although supplemental Figure 8—figure supplement 1 is titled "Statistical evaluation of the predictive model for future cell behavior in *C. elegans* and zebrafish and a schematic describing the method used for statistical simulations", it is not apparent that zebrafish data were used to generate the model or that the model was used to predict cell cycling in zebrafish cells. To what degree can this model predict (or be modified to enable prediction of) quiescence or cycling of cells within other *C. elegans* tissues? In zebrafish?

---

## [Author Response]

Essential revisions:1) Throughout the manuscript, the authors conflate being predictive with being determinative, as demonstrated in the legend of Figure 8: "CDK activity shortly after mitotic exit determine future cell fate". Even if the correlation between CDK^inc^ / continued cycling and CDK^low^ / quiescence is 100%, it does not necessarily mean that CKI/CDK activity is instructive (i.e. determinative) for entry into quiescence. CKI activity may not even be required, as there could be other, redundant inputs into cell cycle exit. For this reason, we recommend re-wording some results and conclusions, taking care not to be too strident in concluding mechanism of quiescence and "cell fate" from the reporter analyses.

We have reworded the text in the manuscript, and in particular the figure legend of Figure 8, to clarify that levels of CDK activity in cells that have just exited mitosis is *predictive* of whether a cell will divide or not.

2) The authors often equate quiescence after cell cycle exit with terminal differentiation, as in the first sentence of the conclusion: "[the sensor] can distinguish between proliferative and terminally differentiated cells within an hour of cell birth". The sensor is highly predictive of exit from the cell cycle and thus likely a quiescent state, but not necessarily terminal differentiation, as cells in many different species can be quiescent without terminally differentiating. Again, the reviewers recommend simply taking care with the language and sticking close to what the sensor is actually measuring, which will in no way diminish the value of the tools or the study. We also recommend changing the title of this study to reflect this distinction, from "proliferation-terminal differentiation decision" to "proliferation-quiescence decision", or something similar.

We have toggled back and forth between proliferation and various distinct states of G0, such as quiescence and terminal differentiation. Thus, we welcome this recommendation by the reviewers. We have modified the manuscript to reflect our change in language, and we now use quiescence as opposed to terminal differentiation to describe cells that have entered G0.

3) On a related note, the finding that the vast majority of zebrafish tail bud cells at 24 hpf are "terminal" is surprising, and seemingly at odds with the many tail bud cells in Figure 7A with high cytoplasmic DHB levels and presumed to be in G2, and with supplementary Figure 7—figure supplement 1D, where 50% of cells are shown to be in G1.

We observed cells in the tailbud at all phases of the cell cycle (Figure 7B), and cells in G1 will likely go on to divide and are not in fact terminal. The previous cut off for the palbociclib data was set at 0.60. We changed this cut off to the mean for DHB-mNG based on time-lapse data for G1 which is 0.69. This changed the percentage of cells in G1 in the tailbud very slightly, 51% of cells are in G1 in the control whereas 87% are in G1 when treated with palbociclib. If you look at Figure 7A and Figure 7—figure supplement 1B zoom outs, you will see cells with all localizations of the sensor – while the zoom in shows regions that are enriched for presumed G2 cells.

4) In zebrafish embryos/larvae, the authors compare DHB ratios in progenitor cells at 24 hpf and with differentiated cells at a significantly later time-point: 72 hpf (Figure 7). When and how often are transgenic zebrafish embryos heat-shocked to induce DHB sensor expression? If embryos examined at 24 hpf and 72 hpf were heat-shocked at the same point in development, could reduced cytoplasmic levels of DHB at 72 hpf compared with 24 hpf be explained by gradual degradation of the DHB sensor? To ensure that differences in DHB ratios reflect the distinction between G1 and G0 rather than differences in developmental age, we recommend that these comparisons be made between differentiated and undifferentiated cells within embryos of the same stage.

The animals are heat shocked 3-5 hours prior to imaging so the total amount of expression should be comparable, however since the sensor is ratiometric the total amount of expression is not relevant and any decay over time would not result in a change in the determination of the cell cycle state as long as signal is still present. However, we agree that it would be beneficial to compare animals at the same developmental stage and thus we have changed Figure 7 to compare the adaxial cells of the posterior-most somites, which are the first cells to differentiate in the somites, with the notochord progenitors, both at 24 hpf, in addition to the comparisons made to the differentiated cells at 72 hpf. Based on the published literature, adaxial cells are slow muscle precursors and are considered to be in a quiescent state through the cooperative action of Cdkn1ca (p57) and MyoD (Osborn et al., 2011). The forming somites are known to be highly proliferative (Mendieta‐Serrano et al. 2013), and based on previous FUCCI data, it is known that the notochord cells cycle leave G1/G0 in the more anterior forming notochord following convergent extension (Sugiyama et al. 2014, Sugiyama et al. 2014).

These comparisons are now shown as micrographs in Figure 7F (posterior somite), 7G (24 hpf adaxial cells), and 7I (24 hpf NPCs) and quantified in Figure 7K. In support of our argument that we can distinguish between CDK^inc^(G1) and CDK^low^(G0) cell populations, we detect a significant difference between the mean CDK activity ratio of 24 hpf adaxial cells (0.13), muscle (0.13), and epidermis (0.13) (all considered to be terminally differentiated) as compared to the 24hpf NPCs (0.32).

We have rearranged the original text to specifically addresses this point in subsection “Visualization of Proliferation and Terminal Differentiation Quiescence during Zebrafish Development”:

“…To examine differences between proliferating and quiescent cells, we examined CDK activity in the somites, which are segmental mesodermal structures that give rise to skeletal muscle cells and other cell types (Martin, 2016), and adaxial cells, cells positioned at the medial edge of the somite next to the axial mesoderm (Figure 7F, 7G). The adaxial cells are the slow muscle precursors and are considered to be in a quiescent state through the cooperative action of Cdkn1ca (p57) and Myod (Osborn et al., 2011)…**”**

“We next sought to determine if we can differentiate between the G1 and G0 state based on ratiometric quantification of DHB. We compared adaxial cells to notochord progenitor cells, which are held transiently in G1/G0 before re-entering the cell cycle upon joining the notochord (Sugiyama et al., 2014; Sugiyama et al., 2009) (Figure 7I). Notably, the mean DHB-mNG ratio of the notochord progenitors (0.32±0.08) is significantly higher than the DHB-mNG ratio of the quiescent adaxial cells (0.13±0.04; Figure 7F, 7J). This elevated DHB ratio is consistent in notochord progenitors at two other earlier developmental stages, 90% epiboly (0.28±0.08) and 18 somites (0.27±0.09; Figure 7K). The mean DHB-mSc ratio in the notochord progenitors (0.33±0.08) is also significantly different than the differentiated epidermis (0.13±0.03; Figure 7J). Based on this difference in DHB ratios between notochord progenitors and differentiated cell types, including muscle and epidermis (Figure 7K), and our knowledge of the normal biology of these cells, we conclude that the CDK sensor can infer cell cycle state in the zebrafish, as it can distinguish between a cycling G1 state and a quiescent G0 state.”

5) In *C. elegans*, the authors showed that each tissue exhibits a distinct range of DHB ratios throughout the cell cycle. In other words, that DHB ratios cannot necessarily be directly compared between different tissues. However, comparisons in zebrafish between epidermal cells at 72 hpf and notochord progenitor cells (Figure 7H) are used to illustrate the differences in DHB sensor ratios between G1 and G0. Given the tissue-specific nature of the DHB ratios, these data would be more convincing if sensor levels were compared between cells of the same tissue type (for example, notochord progenitors with differentiated notochord.)

See below.

6) The predictive model appears to have been trained using the VPC lineage and is shown to predict behavior only of VPC cells. Although supplemental Figure 8—figure supplement 1 is titled "Statistical evaluation of the predictive model for future cell behavior in *C. elegans* and zebrafish and a schematic describing the method used for statistical simulations", it is not apparent that zebrafish data were used to generate the model or that the model was used to predict cell cycling in zebrafish cells. To what degree can this model predict (or be modified to enable prediction of) quiescence or cycling of cells within other *C. elegans* tissues? In zebrafish?

We have split out the response to comment 5 and 6 as we think they are closely related.

We agree with the reviewers that within *C. elegans* we see a different range of CDK activity between cell types, but we believe that it is still possible to compare G1 vs. G0 at mitotic exit. In support of this, we trained a new classifier using 75% of all collected trace data from *rps-27>DHB::GFP* across all three lineages in *C. elegans*: the SM, uterine, and VPCs. We then used the remaining 25% as “test” data, and the model predicted correctly the difference between a CDK^inc^/proliferative cell and a CDK^low^/quiescent cell 93% of the time. Specifically, in the prediction proliferative cells were called correctly 34/38 (89%) of the time and quiescent cells were called correctly 28/29 (97%) of the time. The overall prediction was correct 62/67 (93%) of the time.

This data is presented in in subsection “A Bifurcation in CDK Activity at Mitosis is Conserved in *C. elegans* and Zebrafish” as:

“Finally, to determine whether we could predict future cell behavior independent of cell type, we trained a new classifier using 75% of all collected *C. elegans* timelapse trace data from the SM, uterine and VPC lineages. We used the remaining 25% of traces as test data. When cross-referenced with the known *C. elegans* lineage, our cell-type agnostic classifier correctly predicted the difference between a CDK^inc^ proliferative cell and a CDK^low^ quiescent cell 93% (62/67) of the time.”

…However, comparisons in zebrafish between epidermal cells at 72 hpf and notochord progenitor cells (Figure 7H) are used to illustrate the differences in DHB sensor ratios between G1 and G0. Given the tissue-specific nature of the DHB ratios, these data would be more convincing if sensor levels were compared between cells of the same tissue type (for example, notochord progenitors with differentiated notochord.)

Please see our response to this question above (in reference to comment #4).

…it is not apparent that zebrafish data were used to generate the model or that the model was used to predict cell cycling in zebrafish cells. To what degree can this model predict (or be modified to enable prediction of) quiescence or cycling of cells within other *C. elegans* tissues? In zebrafish?

While we were able to demonstrate with a new classifier (see above) that we can compare CDK activity after anaphase across tissue/cell type in *C. elegans*, unfortunately, the 66 traces of cell births we collected using the mScarlet:DHB transgene in zebrafish was not sufficient to generate a classifier. While we do believe that if enough data was collected and quantified (the real bottleneck at this point is the manual data quantification in complex 4D datasets), that it would be possible to build a classifier, and it is something we would like to do in the future. But due to these limitations, we have removed any mention of zebrafish in reference to an ability to be predictive based on DHB ratios. However, we have demonstrated that there are significant differences between traces of cells that are born into CDK^inc^ vs. CDK^low^ states in our fish data, just as we see in our *C. elegans* data, and that this corresponds with what we know about the cell biology of different tissues.